# Combination of Enzymes and Deep Eutectic Solvents as Powerful Toolbox for Organic Synthesis

**DOI:** 10.3390/molecules28020516

**Published:** 2023-01-05

**Authors:** Davide Arnodo, Elia Maffeis, Francesco Marra, Stefano Nejrotti, Cristina Prandi

**Affiliations:** Dipartimento di Chimica, Università degli Studi di Torino, Via P. Giuria 7, I-10125 Torino, Italy

**Keywords:** deep eutectic solvents, biocatalysis, sustainable processes, asymmetric synthesis

## Abstract

During the last decade, a wide spectrum of applications and advantages in the use of deep eutectic solvents for promoting organic reactions has been well established among the scientific community. Among these synthetic methodologies, in recent years, various examples of biocatalyzed processes have been reported, making use of eutectic mixtures as reaction media, as an improvement in terms of selectivity and sustainability. This review aims to show the newly reported protocols in the field, subdivided by reaction class as a ‘toolbox’ guide for organic synthesis.

## 1. Introduction

Biocatalysis is nowadays considered as a green and sustainable technology for transformation processes. Water and phosphate buffers have been employed for years as solvents for biocatalyzed reactions; however, drawbacks related to the low solubility of organic compounds in such media have deeply limited the applications of biocatalysts in organic synthesis. An ideal solvent for biotransformation should be nontoxic, biocompatible, biodegradable, and sustainable. In addition, it needs to support high enzyme activity and stability. Enzymes have been demonstrated to be active and stable in a non-aqueous media. In the literature, the use of hydrophobic organic solvents, supercritical fluids, and ionic liquids (ILs), has been successfully reported. Since the seminal report by Verpoorte [1], NADES (natural deep eutectic solvents) are well placed to overcome some of the main limitations encountered in the use of “non-natural” solvents. DES are generally described as low melting mixtures of HBD (hydrogen-bond donor) and HBA (hydrogen-bond acceptor) components embedded in a tight network of hydrogen bonds. When the compounds that constitute the DESs are primary metabolites, namely amino acids, organic acids, sugars, or choline derivatives, they can provide a cytoplasm-like natural environment, meaning that enzymes can transform unnatural substrates in a natural environment [1]. DESs are sometimes related to ILs due to their similarity in some properties; however, the components in DES are not necessarily charged compounds, moreover the class includes an increasing number of mixtures with a wide range of physicochemical properties. The use of NADESs for an enzyme-catalysed reaction could be then considered as a chance to perform the reaction in an environment close to intra or extracellular physiological conditions. Due to the growing interest in biocatalytic reactions run in DES, in recent years, a few exhaustive reviews covering this topic appeared in the literature [2,3,4,5,6,7]. As a complementary contribution to the field, this review lays out the latest updates (2010–2022) on the use of DES in biocatalytic reactions, with the purpose of assembling an overview of synthetic transformations realized with enzymes in NADES, classified according to an IFG (interconversion of functional groups) approach with the aim of providing organic chemists with a toolbox of new and greener alternatives to traditional ones. The review considers reductions, oxidations, hydrolyses, esterifications and transesterifications and an additional chapter dedicated to all those transformations not included in the previous ones. Each paragraph is organized in two sections, one for reactions performed by isolated or immobilized enzymes and one dedicated to whole cell conversions. Indeed, the use of whole cells guarantees the advantage to provide a natural environment and cofactor regeneration; moreover, it prevents enzymes from their denaturation and inactivation which may occur under harsh conditions and unconventional media. Without the requirement for enzyme purification and cofactor addition, whole-cell catalysts also represent the cheapest form of catalyst formulation. Throughout the review, particular attention is also dedicated to the enantioselectivity of the considered transformations, since it represents a notable feature of enzymatic catalysis. The induction of chirality from the enzyme to the product of the reaction occurs thanks to the spatial arrangement of the functional groups in the active site, which is strictly related to the conformation adopted by the protein. Since such conformations are affected by the interaction with the solvent, it is quite likely that the enantiomeric excess of a biocatalytic reaction could be altered, either in a positive or negative way, by changing the environment from “classic” conditions, such as aqueous medium (phosphate buffer), to a more structured DES-like system.

## 2. Reductions

Redox reactions are performed by three main categories of enzymes: oxygenases, oxidases and dehydrogenases. Among them, alcohol dehydrogenases—also termed carbonyl reductases—have been widely used for the reduction of carbonyl groups (aldehydes, ketones) and therefore reviewed in this paragraph.

In general, redox enzymes require co-factors that are essentially co-substrates: they donate or accept the chemical equivalents for reduction/oxidation emerging from the enzymatic reaction in an altered form. In enzyme-catalyzed synthesis and due to their high cost, co-factors must either be used in stoichiometric amounts or be regenerated in situ by a separate reaction [8,9]. The most common cofactors are usually nicotinamide adenine dinucleotide (phosphate) (NADPH) and, less frequently, riboflavin-5′-phosphate (FMN), flavin adenine dinucleotide (FAD) and pyrroloquinoline quinone (PQQ) [10].

Instead, in intact cells, the regeneration of redox cofactors is governed by cellular metabolism. For this reason, biocatalysis by means of whole-cells is rapidly growing and widespread for redox reactions. Unfortunately, whole-cell systems have some drawbacks, for example, product toxicity, byproduct formation, poor substrate uptake rates and tricky products isolation [11].

### 2.1. Reductions with Isolated Enzymes

Cicco et al. in 2018 [12], reported the first application of purified ketoreductases (KREDs) in the asymmetric bioreduction of ketones using deep eutectic solvents (DESs, Table 1, entry 1). They noticed that in DES-buffer mixture, the performance of the biocatalyst was enhanced: the increase of DES concentration led to a higher enantioselectivity (ee) of the resulting secondary alcohol. The best results in terms of conversion (>99%) and *ee* (>99%) of the corresponding secondary alcohols were obtained using choline chloride/glycerol (ChCl/Gly, molar ratio 1:2) or ChCl/sorbitol 1:1 and phosphate buffer; DES content ranging from 50% to 20% (*w*/*w*). Finally, the authors propose to combine a metal-catalyzed isomerization reaction of allylic alcohols with the previously described enantioselective bioreduction in aqueous buffer eutectic mixtures both in a sequential and in a concurrent strategy, thus describing the first example of a one-pot chemoenzymatic cascade. Another interesting catalyzed reduction in DESs was explored in 2020 by Chanquia et al. [13], using alcohol dehydrogenase (ADH, Table 1, entry 2). For economic reasons, cofactors are best used in catalytic amounts and combined with an in-situ regeneration system, which normally implies a second enzymatic reaction, usually mediated by glucose dehydrogenase (GDH) or formate dehydrogenase (FDH) together with an adequate ancillary co-substrate. The authors selected ADH from horse liver (HLADH), which can accept a broad range of diols as substrates and catalyzes reactions in DES-aqueous systems. As reaction media, a DES consisting of ChCl and 1,4-butanediol (1,4-BD) was chosen, in combination with buffer (Tris-HCl, 20% *v*/*v*). Acting as a smart co-substrate, 1,4-BD shifts the equilibrium of the reduction when oxidized to γ-butyrolactone (GBL), a thermodynamically stable and kinetically inert coproduct. Of relevance is the production of cinnamyl alcohol, for which proof-of-concept productivities (~75 g L^−1^ d^−1^) are highly promising. The same enzyme was investigated by Bittner et al. in 2022 [14], studying the influence of different DES and their individual components on HLADH. The best result was obtained using ChCl/Gly 1:9 with 20% *v*/*v* of water (Table 1, entry 3). The great amount of glycerol was used as strong stabilizer for the enzyme, while ChCl is detrimental. Instead, water was needed to diminish the viscosity of the mixture. The enzyme is highly active and stable in these conditions, transforming cinnamaldehyde to cinnamyl alcohol with a productivity of 15.3 g L^−1^ d^−1^. As already mentioned, 1,4-BD it is a smart co-substrate in these reactions because it turns in GBL which is an inert co-product, thus shifting the equilibrium. Moreover, HLADH was also used by Meyer et al. in 2022 [15] in a simple and effective system called the Thermomorphic Multiphasic System (TMS, Table 1, entry 4). The latter behaves differently in relation to the temperature: above and below certain temperatures the system is biphasic, and in between this range the system is monophasic. The TMS is composed by a DES, prepared by mixing lidocaine and oleic acid (molar ratio 1:1) and an aqueous potassium phosphate buffer solution. The great advantage of this methodology is based on the easy separation of the product and recovery of the enzyme (to be used in a following catalytic cycle, up to three times) by simply switching to a biphasic system at the end of the reaction.

### 2.2. Whole-Cells Catalysed Reductions

Panić et al. in 2018[16] enantioselectively synthesized chiral molecules using whole-cell biocatalysis (plant cells) in combination with natural deep eutectic solvents (NADES Table 2, entry 1). In detail, the reduction of 1-(3,4-dimethylphenyl)ethanone mediated by carrot roots was successfully conducted in cholinium-based eutectic-mixture–water (NADES content 30–80% *w*/*w*). The authors demonstrated that NADESs are able to influence both the conversion and the *ee* depending on the type of hydrogen bond donor and on the amount of water; the enantioselectivity of the biocatalyst towards 1-(3,4-dimethylphenyl)ethenone was deeply influenced by the solvent, reaching the point of being reversed: the highest conversion was obtained in pure water, obtaining the (*S*)-alcohol with a 95.6% *ee*, while moderate conversions were obtained in NADES–water systems with a maximum enantiomeric excess of 75% in choline chloride/xylose (ChCl:Xyl, 2:1) and water (30% *w*/*w*). Another useful whole-cell reduction, reported by Vitale et al. in 2018 [17], was performed to produce (*S*)-rivastigmine (Table 2, entry 2), a key intermediate involved in the synthesis of an important drug to treat moderate Alzheimer’s dementia. The reaction was carried out by *Lactobacillus reuteri* DSM 20,016 resting cells, able to perform a R-regioselective bioreduction of the aromatic ketone in phosphate buffered saline (PBS). This simple and economical chemoenzymatic process made it possible to obtain (*S*)-rivastigmine in an overall yield of 78% and 98% *ee*. Moreover, the reductions of 3-acetylphenyl-*N*-ethyl-*N*-methylcarbamate and acetophenone were performed using baker’s yeast resting cells in water and choline-chloride-based DES/water to obtain the corresponding secondary alcohols; surprisingly, the obtained products show the opposite stereochemistry. Another improvement obtained using the appropriate DES was disclosed by Li et al. in 2019 [18] (Table 2, entry 3). They designed a novel oligopeptide-based DES containing choline chloride (ChCl) and glutathione (GSH). The reduction of 3,5-bis(trifluoromethyl) acetophenone was achieved using *Trichoderma asperellum* ZJPH0810 in phosphate buffer and the new oligopeptide-based DES as a cosolvent. The best results were obtained increasing the substrate loading by two-fold (vs. aqueous buffer): yield >90% and *ee* > 99%. Furthermore, the asymmetric reduction of other substrates using the ChCl:GSH-containing system have been tested and compared to the aqueous system: the substrate loading was improved (50 vs. 100 mM when catalyzed by *Candida tropicalis* 104), the yield was higher (i.e., from 70 to 88% when catalyzed by *C. tropicalis* 104, from 66 to 84% by *Candida parapsilosis* ZJPH1305 and the reaction time was shortened (24 vs. 30 h when catalyzed by *C. tropicalis* 104, or 1.0 vs. 1.5 h by recombinant *Escherichia coli*). In the same year, Panić et al. [19] used *Saccharomyces cerevisiae* to prepare [(S)-1-(3-methylphenyl)ethanol, (S)-1-(3,4-dimethylphenyl)ethanol and (*S*)-1-(2,4,6-trimethylphenyl)ethanol] (Table 2, entry 4). Choline-chloride-based NADES were prepared by the authors using different amounts of water (from 30 to 80% *v*/*v*) to obtain the best yield and *ee* in the reduction of 1-(3-methylphenyl)ethanone, 1-(3,4-dimethylphenyl)ethanone (DMPA) and 1-(2,4,6-trimethyphenyl)ethanone (vs. water). These are the mentioned NADES: choline chloride/glucose (ChCl/Glu, 1:1), choline chloride/ethylene glycol (ChCl/EG, 1:2) and choline chloride/glycerol (ChCl/Gly, 1:2). DMPA was converted with the highest enantioselectivity using chloride chloride/glycerol (ChCl/Gly) with 30% *v*/*v* of water. Moreover, the asymmetric reduction using DMPA in recycled NADES (ChGly30) was achieved on a preparative scale. More research has been carried out in the area of DESs by He et al. [20] They synthesized validated the bioreduction performances of recombinant *Escherichia coli* cells, using 2-chloro-1-(3,4-difluorophenyl)ethanone (CFPO) as substrate in different combination of choline and amino acids as DESs components (Table 2, entry 5). In choline acetate/lysine (ChAc/Lys), the asymmetric production of (*S*)-2-chloro-1-(3,4-difluorophenyl)ethanol ((S)-CFPL) was realized. The authors pointed out that ChAc/Lys has a role in coenzyme regeneration and, additionally, improves the permeability of the cell membrane, leading to higher conversions. (S)-CFPL (Table 2, entry 5) was synthetized (87% yield and >99% *ee*) starting from 1 mol L^−1^ of CFPO optimizing the reaction in a ChAc/Lys-buffer system. The process was scalable at 500 mL preparative scale, obtaining a satisfactory conversion. The chiral alcohol (*1R*)-1-(3,4-dimethylphenyl)ethanol was produced in NADES starting from the substrate 1-(3,4-dimethylphenyl)ethanone by Pavoković et al. in 2020 [21] (Table 2, entry 6). This reduction was mediated by a plant cell culture (*Beta vulgaris* L. subspecies *vulgaris*, sugar beet). The NADES containing glucose or polyalcohols (glycerol and ethylene glycol) and choline chloride were used in combination with water (30, 50 and 80% *w*/*w*) to assess the conversion and *ee* of the bioreduction; these latter parameters were found to be dependent on the hydrogen bond donor, while the water content was able to modify the enantioselectivity. Moreover, NADES affected the permeabilization of the sugar beet cells, inducing a change in their metabolism. It was also possible to recycle sugar beet cultures after 3–7 days maximum of incubation. However, in the work by Peng et al. [22] the catalytic rate of *Kurthia gibsonii* SC0312 cells was increased by 22%, by only using small concentration of DES (chloride/1,4-butanediol, ChCl/Bd 1:4, 2% in PBS). Under these conditions (*R*)-1-phenyl-1,2-ethanediol in 80% yield and >99% *ee* was achieved (Table 2, entry 7). Shifting to recombinant *E. coli*, Xia et al. [23] performed the reduction of 1-(4-(trifluoromethyl)phenyl)ethan-1-one in betaine/Lys medium obtaining a remarkable yield of 92% for (*R*)-1-[4-(trifluoromethyl)phenyl]ethanol and enantioselectivity > 99.9% (Table 2, entry 8), compared to a 78% yield in phosphate buffer system. This bioprocess was also feasible at 500 mL preparation scale. Moreover, the same substrate was reduced to the corresponding *R*-alcohol in Bet/Lys-containing system (1% *w*/*v* of DES) by *G. geotrichum* ZJPH1810 whole cells in 78% yield and 64% *ee*; the same biocatalyst was employed to reduce 2,6-dichloro3-fluoroacetophenone in L-Carnitine/Lys containing system (1% *w*/*v* of DES), among others, obtaining (*S*)-1-(2,6-dichloro-3-fluorophenyl)ethan-1-ol in 71% yield and 99% *ee* (Table 2, entry 8). Another interesting publication regards the aqueous bioreduction of 1-(3,5-bis(trifluoromethyl)phenyl)ethan-1-one into the corresponding *S*-alcohol (Table 2, entry 9) disclosed by Bi et al. in 2021 [24]. In this case, the yeast *Cyberlindnera saturnus* ZJPH1807 was used along with L-carnitine/lysine (C/Lys, molar ratio 1:2) NADES and a surfactant. The role of NADES is to improve the permeability of the membrane, the major limitation in aqueous buffer, while the surfactant Tween-80 increased the solubility of the substrate. In these conditions, 81% yield was obtained within 24 h and a substrate loading of 500 mM. The process is scalable to 500 mL. The idea of using a surfactant improves the feasibility of whole-cell biocatalysis with hydrophobic substrates.

The whole-cell biotransformation of 1-(2-(trifluoromethyl)phenyl)ethan-1-one to the corresponding *S*-alcohol, by a strain of *Geotrichum silvicola* named ZJPH1811 (Table 2, entry 10), was performed by Xiong et al. in 2021 [25]. Compared to the buffer system, the key chiral intermediate of Plk1 inhibitor (*S*)-1-[2-(trifluoromethyl)phenyl]ethan-1-ol was produced with an improved yield using choline acetate/cysteine (ChAc/Cys, 1:1) as a cosolvent. Moreover, the addition of methylated-β-cyclodextrin (MCD) permitted to further improve the yield of the process and to demonstrate the synergistic effect of DES and CDs in biocatalysis. Finally, Peng et al. in 2022 [26] performed a dehydration of D-fructose in DES composed by betaine/benzenesulfonic acid (Bet/BSA, molar ratio 1:2) to produce 5-(hydroxymethyl)furan-2-carbaldehyde. This one was used as substrate by recombinant *E. coli* DCF (containing both the reductase and the formate dehydrogenase) to obtain furan-2,5-diyldimethanol (Table 2, entry 11). The whole-cell reaction was carried out at 40 °C and pH 7.5 for 48 h, with a yield of 99.6 ± 0.1% using the same DES, to which water was added.

## 3. Oxidations

Biooxidations represent a very important class of reactions mainly because of the importance of the oxidated products as building block for polymers or bioactive chemicals. Generally, enzymes for oxidative reactions present a metallic core (typically iron or copper-based) which interacts with oxygen atoms and promote their transfer to the substrate.

In this section, a variety of biocatalysed oxidations will be reported, all of them carried out in deep eutectic solvents to ensure green conditions and a good phase transfer taking advantage of the organic nature of these solvents.

### 3.1. Oxidationss with Isolated Enzymes

The most common enzymes used for these reactions are laccases, dehydrogenases, hydroxylases, peroxygenases and oxidases. The main class of substrates are steroids for their significant role in therapeutical applications. Some examples of enzymatic oxidations are summarized in Table 3.

#### 3.1.1. Oxidations Catalyzed by Laccases

Laccases belong to copper-containing oxidases enzymes and catalyze the single-electron oxidation of organic and inorganic compounds with the concomitant reduction of molecular oxygen in water [27]. In this context, Tavares et al. [28] investigated the use of DESs as cosolvents in oxidative reactions catalyzed by laccases. The group of Tavares evaluated the effect of various cholinium- or betaine-based DESs as cosolvents in aqueous solutions on the laccase activity under different conditions. Among them, choline dihydrogen citrate/xylitol (ChDHC/Xyl 2:1) at 25% (*w*/*w*) has been highlighted as the most promising, with an enhancement in the laccase activity up to 200% at 25 °C. In addition, it was shown to be a better storage media at −80 °C, ensuring a preservation of the enzymatic activity up to 125% for 20 days.

The catalytic activity of laccases can be also expanded to the use of redox mediators (*enzyme enhancers*) whose oxidation leads to the formation of highly reactive intermediates which can oxidase the target compound at a high rate. In this direction, in 2021, Yaropolov et al. [27] performed the oxidative polymerization of dihydroquercitin (DHQ), a natural flavonoid, to its corresponding oligomer in a DES-buffer mixture and with (2,2,6,6-tetramethylpiperidin-1-yl)oxyl radical (TEMPO) as enzyme enhancer (Table 3, Entry 1). This reaction is carried out at room temperature, using betaine/glycerol 1:2 as DES 60% (*v*/*v*) and 0.61 mol% of TEMPO. This mixture enabled a high loading of DHQ (>17 mM) and guaranteed stability of the enzyme for 12 h; in addition, the laccase-mediator system leads to a yield of the product of 58 ± 7% with a number average molecular weight of 1800 g/mol and an average chain length of six monomer units. The mechanism is still a hypothesis, and it can be considered as a two-step process: first, the formation of an alkoxy or aryloxy radical by laccase and then the radical substitution of the hydrogen atom in the aromatic ring with the formation of a new C-O bond. The synthesis of oligomeric derivatives of flavonoids are promising for pharmaceutical use due to their antioxidant, antimutagenic, anticarcinogenic, antiviral properties and longer in vivo life. Polyaniline (PANI) can be synthetized by enzymatic oxidation of aniline using laccase (Table 1, Entry 2). PANI is an object of great interest due to its distinctive electrical and optical properties combined with ease of production, low cost of the monomer and environmental stability; among its three oxidation states, the half-oxidate one (emeraldine base, EB) is the most useful since it shows electroconductive property when doped with an acid, to form emeraldine salt (ES). The main difficulty of this method is to control the production of the desired form of the polymer and to avoid over-oxidation. Takaç et al. [29] examined the use of DES D,L-menthol/decanoic acid (Ment/DA) 1:1 as template for the polymerization of aniline, in order to synthesize PANI in the desired form. The reaction has been carried out with 25 mM aniline, 0.73 U mL^−1^ of laccase, 10% of DES at pH = 3 and 15 °C for 24 h with a yield of 84%; *p*-toluensulfonic acid is also added during polymerization as an acid doping agent, to provide PANI with conductive characteristics. In addition, copolymers of aniline can be prepared using laccase, as reported by Yaropolov et al. [30]. Betaine/glycerol 1:2 was chosen as cosolvent in a volume ratio with buffer of 60/40 (% *v*/*v*) which allowed preservation of about 60% of enzymatic activity after incubation for 120 h. In this case, air oxygen is utilized as oxidizing agent and sodium polystyrene sulfonate (PSS) as template.

#### 3.1.2. Oxidations Catalyzed by Peroxygenases

Peroxygenases are a new class of heme-thiolate dependent enzymes catalyzing oxidative reactions using H_2_O_2_ as stoichiometric oxidant [31]. However, if applied in surplus, hydrogen peroxide could lead to oxidative inactivation of the enzyme; thus, it is useful to associate an efficient method to balance the peroxide concentration. In this context, in 2019, Wang et al. [31] proposed an in-situ H_2_O_2_ generation system based on choline oxidase (ChOx)-catalyzed conversion of choline into betaine with the concomitant generation of two equivalents of hydrogen peroxide to drive peroxygenase-catalysed oxyfunctionalization. In this way, they used ChCl-based NADES both as solvent and as sacrificial electron donor, for the bienzymatic hydrolyzation of ethylbenzene and, at the same time, epoxidation of cis-β-methylstyrene (Table 3, Entry 3). For this reaction, choline oxidase from *Arthrobacter nicotianae* (AnChOx) and the recombinant peroxygenase from *Agrocybe aegerita* (rAaeUPO) have been used. The hydroxylation has been carried out in ChCl/Urea/Gly 1:1:1 75 (% *v*/*v*) as a solvent with a yield of 84% while the epoxidation has been carried out in ChCl/Pro/H_2_O 1:1:1 50 (% *v*/*v*) with a yield of 35%; both the reactions were run at 30 °C, at pH = 7 and for 24 h as optimized reactions conditions. This procedure demonstrated to be a good sustainable route for oxidizing different substrates, such as the enantioselective sulfoxidation of thioanisole (Table 3, Entry 4)[32] in ChCl/Urea 1:2 50 (% *v*/*v*) in water buffer, in alkaline media, at 30 °C for 24 h. Despite conversions in this system never exceeding 50–60%, (*R*)-sulfoxide was obtained with an >99% *ee*; moreover, the corresponding overoxidation product was never observed. These results are very promising because they propose an efficient, enantioselective, and green method to synthetize chiral sulfoxides which are widely found in drugs, chiral intermediates as well as chiral ligands in catalysis. Another example of this synergistic work is reported in 2022 [33], where the dual function of ChCl/Pro/H_2_O 1:1:1 as solvent for the extraction of limonene from waste orange peels and for its biocatalytic oxidation as well as co-substrate (Table 3, Entry 5) was investigated. In this way, carvol and carvone are successfully obtained through a cascade reaction sequence.

#### 3.1.3. Oxidations Catalyzed by Peroxidases

Peroxidases are a large group of oxidoreductases which commonly break up peroxides to generally promote oxidative radical polymerization of the substrates. In this context, horseradish peroxidase (HRP) is a heme-containing enzyme which is regularly utilized to catalyze the oxidation of a variety of organic compounds in the presence of hydrogen peroxide [34]. In 2014, Yang et al. [34] investigated the activity and stability of HRP in 24 different DESs composed by ChCl or ChAc with four hydrogen bond donors (urea, glycerol, acetamide and ethylene glycol). They demonstrated that, for the DESs composed by the same salt and an HBD, the enzyme activity appeared to be higher with a higher salt/HBD ratio and that ChCl-based DESs have been found to be superior in terms of promoting HRP performance. Particularly, ChCl/U 1:2 was tested in the catalytic removal of phenols in industrial wastewaters: the addition of the DES accelerated the process with a higher rate obtained at a higher concentration, with a reduction of the treatment time from 20.6 min to 14.9 min at a concentration of 100 mM of DES. In the end, all 24 DESs investigated have shown to highly stabilize the HRP structure. Peroxidases are regularly utilized also to induce polymerization of vinyl monomers. In particular, Mota-Morales et al. [35] reported the HRP-mediated free radical polymerization of acrylamide in ChCl/U 1:2 and ChCl/Gly 1:2 (Table 3, Entry 6). Initiation by free-radical species was generated by the catalytic system HRP/H_2_O_2_/2,4-pentanodione at 50 °C. DESs concentrations were fixed to 80% (*v*/*v*) to preserve the HBD supramolecular complexes structure and to lower the viscosity of these solvents thus facilitating the diffusion and mass transfer in the reaction media. Polymerization was successfully achieved with very high yields in all cases (90% for ChCl/U and 99% for ChCl/Gly); ChCl/U allowed the synthesis of PAA with similar M_w_ and narrower polydispersity (PDI) to the corresponding experiment in water while ChCl/Gly did not offer any benefits in terms of M_w_ and PDI. Despite these results, taking advantage of the low freezing point of ChCl/Gly in its pure state (−40 °C), polymerization was also tested at 4 °C in this solvent and PAA with a higher M_w_ than that at 50 °C was obtained. In addition, HRP in ChCl/U has shown a higher thermostability thanks to hydrogen bonds established with the surface residues of the enzyme which forms a rigid structure and promote the protein stability.

#### 3.1.4. Oxidations Catalyzed by Cytochromes

Cytochrome enzymes are very common in many different tissues of human body such as intestinal and liver tissues. They are efficient biomarkers in environmental toxicology researches due to their outstanding sensitivity and inducibility; they also promote oxidations of various steroids, drugs cancer-causing toxins, liposoluble vitamins and natural products [36]. Among them, P450 BM-3 is a NADPH-dependent fatty acid hydroxylase enzyme, and it is an effective catalyst in the production of indigo from indole, an alkaloid insoluble in water, ethanol and diethyl ether, used as dye in textile, food, and pharmaceutical industries. In this context, in 2020, Yao et al. [36] proposed a version of this synthesis in NaDES (Table 3, Entry 7) to solve the main issues of the industrial process: generation of a great amount of pollutants and subsequent damage to the environment. The reaction has been carried out in a two-phase system of two different alcohol-based DESs, ChCl/EG 1:2 and ChCl/PG 1:2, and aqueous buffer with similar results; the optimal conditions consist in 0–1 mmol/L of indole, pH = 7 and temperature of 25–35 °C. The enzyme exhibited higher thermal stability and activity in this system than in the buffer or in common organic solvents due to the presence of DES whose hydrogen-bond network protects and stabilize enzyme’s structure.

#### 3.1.5. Oxidations Catalyzed by Other Enzymes

In addition to these conventional oxidizing enzymes, during the last 10 years, other biocatalysed oxidation reactions were successfully conducted in deep eutectic solvents. Moreover, the use of DES cannot only be limited as solvent, but it can also be utilized for the purification of the final product. In this context, Li et al. [37] developed a new enzyme toolbox for the selective oxidation of 5-hydroxymethylfurfural (HMF) to 2,5-diformylfuran (DFF), 5-hydroxymethyl-2-furancarboxylic acid (HMFCA), 5-formyl-2-furancarboxylic acid (FFCA) and 2,5-furandicarboxylic acid (FDCA). In addition, they used ChCl-based DES to separate HMF and DFF and to improve the purity of the latter one (Table 3, Entry 8). DFF was obtained using galactose oxidase (GO) added with catalase and horseradish peroxidase (HRP) to remove byproduct H_2_O_2_ which could lead to an inactivation of the enzyme. The yield was almost quantitative (92%) at 25 °C in deionized water after 96 h. After the oxidation of HMF to DFF, their separation was required and DES ChCl/Gly showed the best results due to its higher affinity toward HBD molecules (e.g., alcohols and phenols) resulting in a great dissolution. After three times extraction, DFF purity significantly increased from 76% to 93%; however, its recovery was still unsatisfactory (55%) and it potentially needed optimization of its DES composition. HMF is a great bio-based chemical thanks to its two active groups, which could be subjected to various modifications, and its oxidized derivatives are versatile building blocks with promising potential application in fuels, polymers, and drugs. In 2019, Wang et al. [38] proposed a promising method to valorize agricultural wastes using deep eutectic solvents: they used NADESs for the extraction of limonene from orange peel wastes, as solvent for its the chemoenzymatic epoxidation and as sacrificial electron donor for the in situ generation of H_2_O_2_ to promote the reaction (Table 3, Entry 9). For this process, carboxylic acid was used only in catalytic amounts to generate the reactive peracid through reaction with hydrogen peroxide in situ catalyzed by lipase; in addition, H_2_O_2_ was regenerated by choline oxidase (ChOx). ChCl/1,2-propanediol/H_2_O 1:1:1 and ChCl/EG 1:1 yielded the highest extraction efficiencies for limonene at 40 °C in 24 h with a great purity. ChOx generated H_2_O_2_ from ChCl/Pro/H_2_O 1:1:1 with success under neutral to slightly alkaline conditions, at temperatures between 30 and 40 °C and with a concentration of NADES between 25% and 75% (*v*/*v*). The spontaneous epoxidation was carried out at 40 °C (as a compromise between enzyme activity and extraction efficiency) and pH = 7 for at least 2 days: first, the monoepoxide was formed, later followed by some accumulation of the diepoxide product. Orange peels are a great source of terpenoids, and, among them, limonene is largely present and a very versatile chemical: for example, its epoxide-derivatives are great building blocks for polymers. In the same year, Fraaije et al. [39] explored different recycling enzymes as fusion partner for two oxidoreductases, Baeyer–Villiger monooxygenase (BVMO) and alcohol dehydrogenase (ADH), which both act on ketone substrates. These different fused enzymes were tested in a variety of DESs to improve the solubility of substrates which is generally low in water. ADH and BVMO are both NADPH-dependent enzymes: the first one catalyzed asymmetric reduction of ketones to (*R*)- or (*S*)-alcohols in excellent enantiomeric excess, the second one catalyzes regio- and enantio-selective transformation of ketones to esters or lactones using molecular oxygen as an oxidizing agent. To recycle NADPH, regenerating coenzymes could be also covalently fused: formate dehydrogenase (FDH), glucose dehydrogenase (GDH) and phosphite dehydrogenase (PTDH). All the possible fused enzymes resulted in good soluble expression as well as being fully functional as self-sufficient biocatalysts; they showed great tolerance to the analyzed DESs, with ChCl/Gly 1:2 and ChCl/Glu 1.5:1 10–30 (% *v*/*v*) as the most efficient cosolvents. Moreover, the same group [40] investigated the effects of different NADESs in the oxidation of a variety of aromatic alcohols catalyzed by 5-hydroxymethilfurfurale oxidase (HMFO) from *Methylovorus* sp., which is a flavin containing oxidase that only requires molecular oxygen as a mild oxidant. The presence of an electron-donating group led to a decrease in enzyme activity while electron-withdrawing groups had a positive effect on HMFO; the enzyme was also able to accept bulkier alcohols as substrates. DESs were added as cosolvents with a concentration of 60% (*v*/*v*) and Glu/Fru/H_2_O 1:1:6 showed the best results at 30 °C and 20 h reaction. Thus, it was examined in the oxidation of HMF (Table 3, Entry 10) which can give two main products: 5-formylfuran-2-carboxylic acid (FFA) and furan-2,5-dicarboxylic acid (FDCA). DES had positive effects on HMFO activity, the best HMF to FDCA conversion (31% after 24 h) was realized with a concentration of 30% (*v*/*v*). These studies demonstrated that sugar-based NADESs improve to a great extent the performance of HMFO, thanks to their stabilizing effects, which increase an enzyme’s thermostability. In 2022, Gladkowski et al. [41] developed an efficient method for the implementation of lipase-mediated Baeyer–Villiger oxidation of α-benzylcyclopentanones in deep eutectic solvents (Entry 11). They chose lipase B from *Candida antarctica* immobilized on acrylic resin and urea hydrogen peroxide (UHP) as oxidant. The best conversion is obtained in the case of the oxidant as a component of DES (minimal DES): indeed, in ChCl/UHP 1:2 at 55 °C, after 3 days, the observed conversion of the ketones to lactones exceeded 92% for all the substrates. These results were compared with those obtained in the traditional ester solvents, but the reaction carried out in DES presented the further advantage to limit the hydrolysis of lactones to their corresponding hydroxy acids. Furthermore, the use of the minimal DES (mDES) provided both a solvent for the reaction and, at the same time, a source of oxidizing agent.

**Table 3 molecules-28-00516-t003:** Oxidations with isolated enzymes.

Entry	Oxidations with Isolated Enzymes	Ref.
1	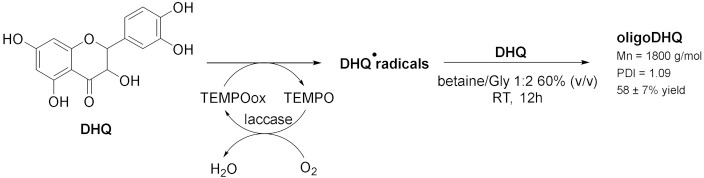	[27]
2	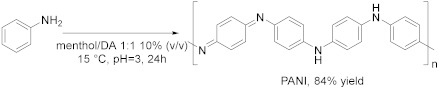	[29]
3	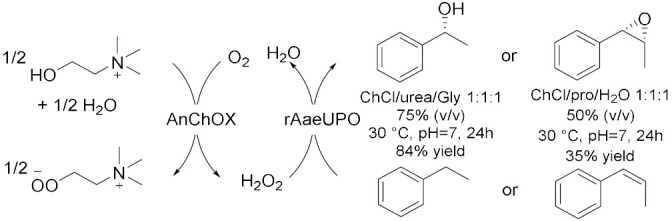	[31]
4	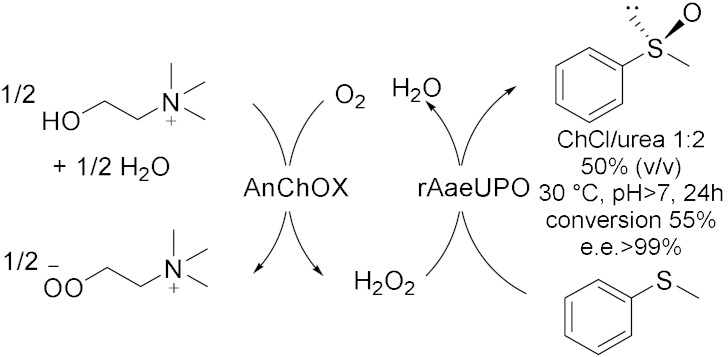	[32]
5	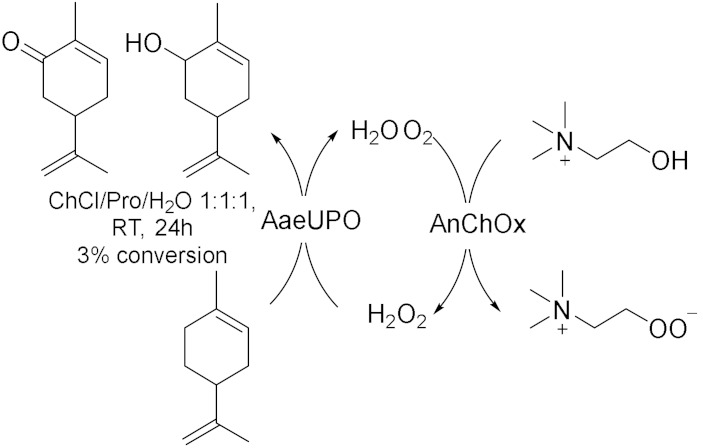	[33]
6	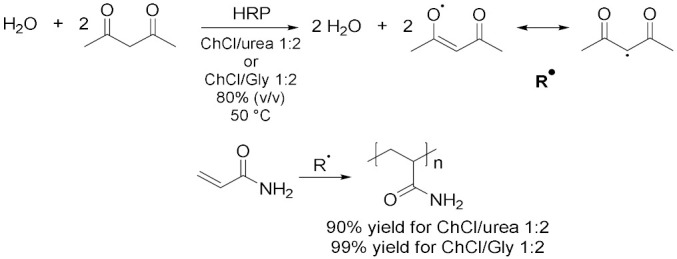	[35]
7	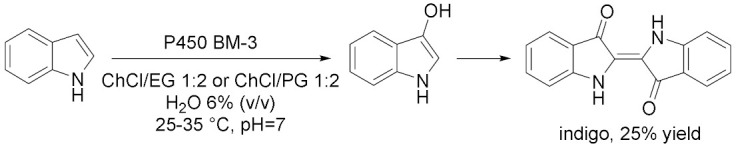	[36]
8	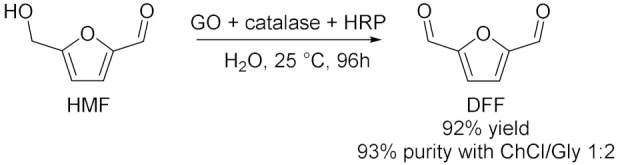	[37]
9	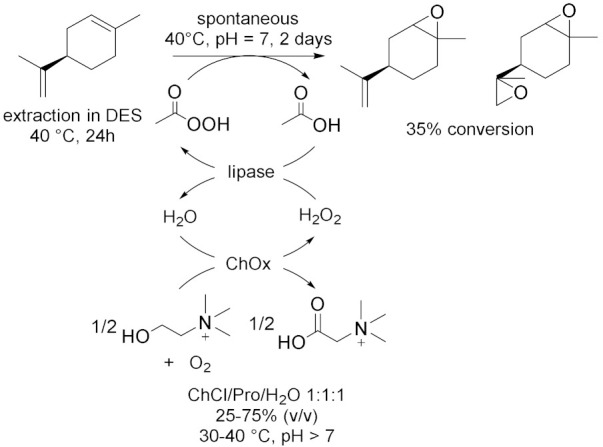	[38]
10	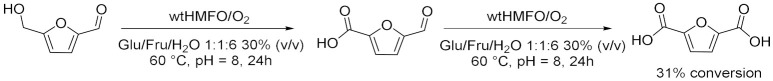	[40]
11	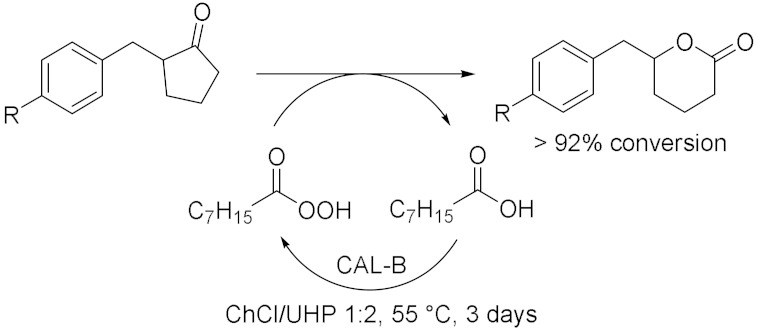	[41]

### 3.2. Whole-Cells Catalysed Oxidations

In Table 4, some examples of whole-cells oxidation reactions are summarized. In all of them, the combination with unconventional media (such as DESs) is proposed, taking advantage of the benefit this class of neoteric solvents have to the conversion performances [42].

Lou et al. [43], in 2014, used ChCl/Gly 1:2 to improve the performance of immobilized *Acetobacter* sp. CCTCC M209061 cells for asymmetric oxidation of 1-(4-methoxyphenyl)ethanol (MOPE) to give enantiopure (*S*)-MOPE, a key chiral building block for the synthesis of cycloalkyl[b]indoles, which are used as treatments n for general allergic responses (Table 4, Entry 1). MOPE is oxidized to 4′-methoxyacetophenone (MOAP) and (*S*)-MOPE is obtained through asymmetric resolution of racemic mixture while NADP^+^ is regenerated thanks to the simultaneous reduction of the co-substrate acetone to isopropanol. The reaction has been carried out in 10% (*v*/*v*) of DES at 30 °C and pH 6.5 with an acetone concentration of 50 mM and a substrate concentration of 55 mM. At these conditions, a conversion of 49% and 99% *ee* have been obtained after 9 h reaction, much higher than in aqueous buffer. The same group extended in 2016 [44] the usage of ChCl/Gly 1:2 to enhance the efficiency of the immobilized whole-cell mediated asymmetric oxidation of MOPE into a [DES][C_4_MIM][PF_6_]/buffer biphasic system at 30 °C, pH 6.5 after 7 h reaction (Table 4, Entry 2). Adding 10% (*v*/*v*) of water-miscible DES into the IL/buffer system has led to an enhancement of the conversion (from 50% to 51%), maintaining the *ee* up to 99%, of the substrate concentration (from 65 to 80 mmol/L) and of the initial rate (from 97.9 to 124.0 μmol/min), shortening the reaction from 10 h to 7 h. Moreover, the relative activity of the immobilized cells remained around 72% after nine batches of successive reuse and the efficient biocatalytic process was feasible up to a 500 mL preparative scale.

The benefic effect of deep eutectic solvents can be extended also as media for the growth and integrity of cells. Lu et al. [45] investigated these effects on *Arthrobacter simplex* cells in the Δ1,2-dehydrogeneration of cortisone acetate (CA) to prednisone acetate (PA), a more effective steroid than its precursor as anti-inflammatory and anti-allergy glucocorticoid agent (Table 4, Entry 3). ChCl/U 1:2 has showed the best performance after 5 h incubation at 30 °C and 4% (*v*/*v*), guaranteeing a minor decrease of membrane integrity (39%) compared with that in a pure aqueous system. In these conditions, a conversion yield of 93% was obtained and, testing the ability to recycle the immobilized cells, only a slight decrease in substrate conversion to 81% after the fifth cycle was observed. This research was later extended [46] with studies about the toxic effects of these solvents. DESs can be non-toxic to the cells in concentrations lower than 40% and water should not be less than 50% (*v*/*v*): in this way, HBDs form additional hydrogen bonds with the molecules of water, and this will not lead to disintegrating effects on cells.

In 2017, Yang et al. [42] proposed the bioconversion of isoeugenol to vanillin catalysed by *Lysinibacillus fusiformis* CGMCC1347 cells (Table 4, Entry 4) scanning 25 DESs and 21 NADESs to discover the most efficient cosolvent. Among the DESs, the addition up to 20% (*v*/*v*), ChAc/U 1:1 and ChAc/EG 1:1 showed the best performances at room temperature and pH = 7. While among the NADES, the best results have been obtained with ChCl/Lac 4:1 and ChCl/raffinose 11:2 at the same conditions. All these cosolvents, have led to an enhancement in the solubility of isoeugenol up to 2.0–2.3 times higher for the DESs and 1.9–2.7 for the NaDESs guarantying an accelerating power on the reaction rate. In addition, the possibility of using of PVA-alginate immobilized cells in 20% (*v*/*v*) of DESs or NaDESs is also investigated: the highest production yields have been obtained with ChCl/Gal 5:2, ChCl/PG 1:1, ChCl/EG 1:1 and the catalytic activity was well maintained for at least 13 cycles (72 h reaction for each cycle) without loss of operational stability. This study offers a promising design for whole-cell biocatalytic synthesis of vanillin, which remains one of the most commonly used flavors in the food, beverage, perfumery, pharmaceutical and medical industries. Years later, Qin et al. [47] proposed the use of DES into an industrial process, in particular, the 15α-hydroxylation of D-ethylgonendione, an important intermediate in the commercially synthesis of Gestodene, a progestin used in menopausal hormone therapy, using fungus *Penicillium raistrickii* (Table 4, Entry 5). The maximum production was obtained with ChCl/Gly 1:2 4% (*v*/*v*) in aqueous system at 28 °C and pH 6.0 after 72 h biotransformation; in these conditions, the conversion was more than 76% which is much higher than the control demonstrating the synergetic effects of the DESs. ChCl/Gly was proved to be the most efficient cosolvent due to its great effect on mass transfer of the substrate and product as well as its better biocompatibility toward the fungus. Steroid hydroxylation is an important object of interest because the activity of hydroxyl steroid at different positions is much higher in various therapies: DESs can improve the catalytic effect of the enzymes and greatly improve the bioconversion efficiency; moreover, whole-cell biotransformation is preferred because the purification of fungal steroid hydroxylase (Cytochrome P450) is very complicated, and NADPH-Cytochrome P450 reductases (CPR) regeneration is also required.

As said before, in the optics of green chemistry, the valorization of natural products represents a very important target. In this context, He et al. [48] proposed a hybrid strategy to synthesize furoic acid from bulrushes via chemoenzymatic route in DES containing citric acid (CA). First, they used ChCl/CA 1:1 in aqueous media (20 vol%) to transform bulrushes to furfural at 180 °C after 30 min with 47% yield; then, they converted furfural in furoic acid with recombinant *E. coli* HMFOMUT whole cells, which present dehydrogenase, at 30 °C for 56 h and with 42% yield (Table 4, Entry 6). This procedure is an environmentally friendly strategy for chemoenzymatic valorization of lignocellulosic biomass via tandem chemocatalysis and biocatalysis in DES. Lignocellulose is an abundant, available, inexpensive, and renewable source of value-added chemicals; furfural is a versatile biobased chemical used in the production of plastics, resins, and polymers; furoic acid is an important intermediate form manufacturing flavors, polymers, fragrances, and pharmaceuticals.

**Table 4 molecules-28-00516-t004:** Whole-cells catalysed oxidations.

Entry	Whole-Cells Catalysed Oxidations	References
1	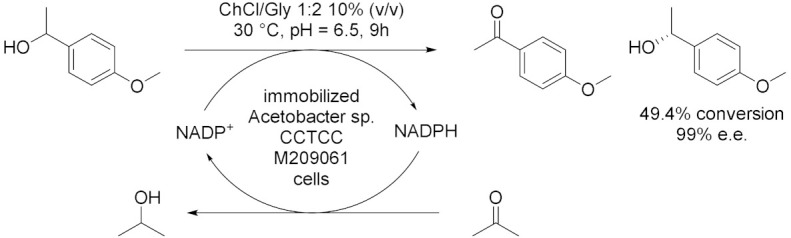	[43]
2	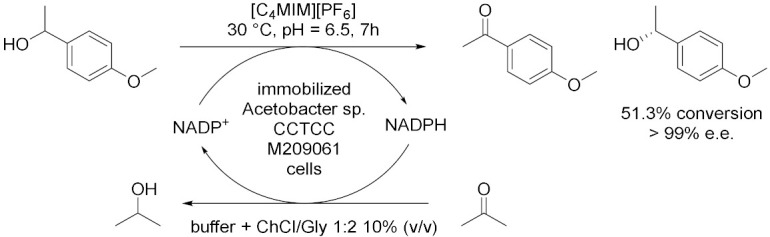	[44]
3	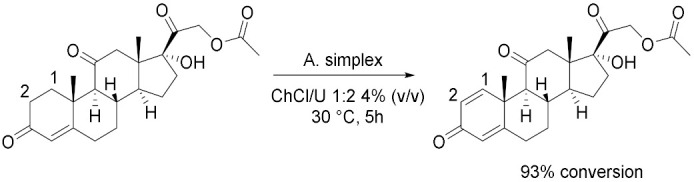	[45]
4	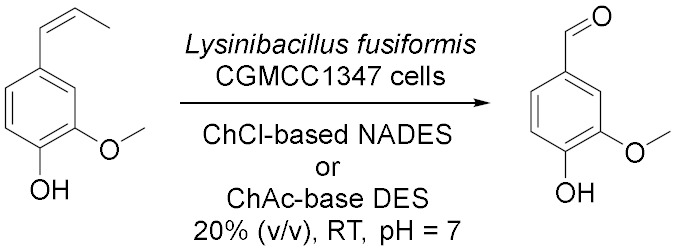	[42]
5	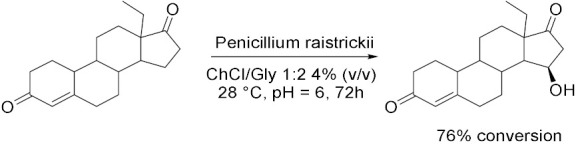	[47]
6	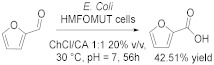	[48]

## 4. Hydrolysis

Hydrolases are the class of enzymes able to catalyze bond cleavage using water and without the usage of cofactors. The natural role of hydrolases is to digest nutrients by cleaving them into smaller molecules and, for this reason, these enzymes accept a large number of substrates. Furthermore, they often have high stereoselectivity and are able to catalyze related reactions (described below) beyond the reversal of hydrolysis [49]. Given their versatility, they represent the dominant type among industrially employed enzymes [50]. The mechanism of action involves a nucleophilic group of a residue in the active site (e.g., the carboxylate group of an aspartic acid in pepsin, the hydroxy group of a serine in serine hydrolases or the thiol of a cysteine in papain) that attacks the corresponding electrophilic substrate to form a covalent acyl–enzyme intermediate [51,52,53]. Then, water or any other nucleophile that can compete with the water (e.g., when working in organic solvents at low water concentration), attacks the acyl–enzyme intermediate, regenerating the enzyme, affording the final product [54]. In recent years, DESs have received more attention in the field of saccharification than in terms of the organic synthesis of molecules.[55] In fact, these are used in the pre-treatment of biomass that are then subjected to enzymatic hydrolysis in order to break the glycosidic bonds.

### 4.1. Hydrolyses with Isolated Enzymes

The work of Cao et al. [56] is positioned halfway between the world of metal–organic framework (MOF) and the biocatalysis. In fact, a nano-/microscale UiO-66-NH_2_ MOF material was prepared and soybean epoxide hydrolase (SEH) was successfully immobilized on it (loading 87.3 mg/g and enzyme activity recovery 88%). The immobilization enhanced pH stability, thermostability, and tolerance to organic solvents compared to free SEH. The immobilized enzyme was stored at 4 °C for 4 weeks and found to retain 97% of its initial activity. Moreover, the enzyme on this support displayed higher substrate affinity and catalytic efficiency compared to free SEH. Finally, SEH@UiO-66-NH_2_ was tested for the asymmetric biocatalysis of 1,2-epoxyoctane which was hydrolyzed into (*R*)-1,2octanediol using DES (Chcl/Urea)-PBS 15% *v*/*v* (yield 41% and *ee* 81%, Table 5, entry 1). Another interesting study by Weiz et al. [57] has to do with the use of a diglycosidase from *Acremonium* sp. DSM24697 (6-*O*-α-rhamnosyl-β-glucosidase) to perform the hydrolysis of hesperidin. This model reaction exploits DESs as cosolvents in order to overcome the low solubility of glycosylated flavonoids in aqueous media (Table 5, entry 2). Deglycosylation takes place using DESs composed by choline chloride and glycerol or ethylene-glycol (molar ratio 1:2) up to 40% (DES-sodium phosphate buffer, *v*/*v*). Moreover, the influence of the single components of DESs was investigated by the authors: glycerol and ethylene-glycol are able to significantly enhance the activity of the enzyme, as opposed to choline chloride, probably because of a concomitant enzyme-mediated transglycosylation reaction. Peng et al. [58] studied the influences of various amounts of DESs on mung bean epoxide hydrolase in phosphate buffer at 35 °C. For example, the addition of 10% *v*/*v* of ChCl/triethylene glycol (1:4) caused an improvement in the enantiopurity of product from 83 ± 1.3% to 88 ± 0.3%. Instead, increasing the amount of DES from 10% to 30% improved the *ee* from 88 to 94% with the drawback of a decreased yield (Table 5, entry 3). The authors pointed out how the immobilization of the enzyme could be essential to improve the stability of the enzyme itself with an increased amount of DES in the system. Years later, another example of hydrolysis was proposed by Xu et al. [59], in which they monitored the activity of a β-glucosidase in DES (ChCl/propylene glycol 1:2)-buffer looking at the formation of *p*-nitrophenol from *p*-nitrophenyl-β-glucopyranoside. The activity of the biocatalyst was increased by 225% in phosphate buffer at pH 6 and 60 °C, performing a pretreatment with 40% *v*/*v* DES, compared to a phosphate buffer system (Table 5, entry 4). Moreover, the effect of pure DES was studied, leading to the conclusion that it deactivates the enzyme, while 6% *v*/*v* of water eliminates this effect, maximizing the kinetic parameters. A different and smart approach, based on the experimental design technique (DOE), was adopted by Yang et al. in 2019 [60]. They performed the hydrolysis of pine nut oil in DES using the biocatalyst amano lipase PS for the preparation of free fatty acids. The hydrolysis was carried out using ChCl/U (1:2) as co-solvent, a water amount of 38%, a DES addition of 43%, a lipase dosage of 7.6% (percentages based on the mass of oil), at 46 °C for 13 h, obtaining the maximum content of free fatty acids in the products of (89%, Table 5, entry 5). Stepping back to the widely used β-glucosidase, a recent work by Han et al. [61] in which they described the hydrolysis of ginsenoside Rb1 into ginsenoside CK in DES (ChCl/EG 2:1)-buffer and through continuous feed technique in order to reduce the inhibitory effects of substrates and products. The best conditions to perform this synthesis are: 30 vol% DES in acetate buffer, pH 5.0, 55 °C, and substrate concentration of 12 mM (productivity equal to 142 mg/L·h, Table 5, entry 6). DES influenced the catalyst by increasing its affinity (49%) and catalytic efficiency (64%) towards intermediate ginsenoside Rd, while the conformation of β-glucosidase was mostly retained. The feed-batch technique permitted to improve the conversion rate by 44% compared to single-batch pure buffer conditions. To top it off, Zang et al. [62] developed a method to prepare quercetin using a rutin degrading enzyme, obtained from germinated Tartary buckwheat, in order to hydrolyze rutin (Table 5, entry 7). The solubility of quercetin in NADESs is enormously higher than water and for this reason they are ideal to extract the final product. The higher extraction efficiency of 291.57 mg/g was achieved using 80% ChCl/Gly 1:1 in water. Precisely in this system, the enzyme also shown its higher degradation rate of 8.36 mg/min·L (pH 7, substrate concentration 1 mg/L, RED 55 uL and 35 °C).

### 4.2. Whole-Cells Catalysed Hydrolyses

Zhang et al. in 2019 [63], used whole cells of *E. coli* BL21-pET21a-rhaB1 treated with ionic liquids (ILs) and DESs for the synthesis of isoquercitrin. The highest catalytic activity was obtained treating cells with ChCl/Urea 1:2 (6% *v*/*v*) in PBS to improve cells’ permeability (Table 6, entry 1). Moreover, the authors demonstrated that whole-cells are more stable than crude rhaB1 in the reaction medium. Under optimized conditions, the yield of isoquercitrin is at about 93% and the cells retained more than 52% of their activity after five uses. A biorefinery for orange peel waste was designed by Panić et al. in 2021 [64]. They investigated the ability of enzymes contained in orange peel to catalyze the enantioselective hydrolysis of (*R,S*)-1-phenylethyl acetate to *R*-phenyethanol using, as a medium, a mixture of 50% NADES (choline chloride/ethylene glycol, molar ratio 1:1) and 50% water (see Tabe 6, entry 2). Not only a high enantioselectivity and yield (83% *ee*, 82% yield) were recorded, but also a stabilizing effect was reported: the hydrolytic enzymes crude extract is stable in the selected NADES for 20 days at 4 °C. Moreover, in this multistep process, NADES plays a wider role because it also made it possible the extraction of polyphenols from impoverished medium. Finally, waste peels were analyzed in order to use them in an anaerobic co-digestion process.

## 5. Esterifications and Transesterification

Esterification and transesterification reactions are some of the most important chemical transformations and found many applications, ranging from the modification of vegetable oils for human consumption to the production of optically pure chemicals [65]. Among the enzymatic processes, lipase-mediated esterifications are used for numerous purposes [66,67,68,69,70,71,72,73,74,75] with the aim of reducing the environmental impact of such processes. These biocatalytic conditions have also been studied to improve reaction parameters such as selectivity, conversion, and by-product formation. In the last years, several bio-based processes involving esterification and transesterification reactions catalyzed by lipases in presence of DESs have been developed (Table 7 and Table 8).

### 5.1. Esterification Reactions with Isolated Enzymes

Bubalo et al. [76] systematically analyzed the advantages and limitations of cholinium-based ILs and DESs used as solvents for immobilized *Candida antarctica* Lipase B (CALB supported over acrylic resin, also known as Novozym 435) [100]. Catalyzed esterification of acetic anhydride with 1-butanol to give butylacetate as product is considered (Table 7, entry 1). Results showed that selected ILs (choline glycinate, choline alaninate, choline asparaginate, choline malate) and DESs (choline chloride mixtures with glycerol, ethylene glycol, and urea as hydrogen bond donors in molar ratio 1:2) are poor media for tested reaction if applied as pure solvents (yield drops below 5%). This happens probably due to the substrate “entrapment” within DES structure through H-bonding. The addition of water (as a protic solvent) to the DESs strongly enhances both enzyme activity and reaction yield. Among different DES/water mixtures tested, the one with EG as hydrogen bond donor at water content of 5% (*w*/*w*) proved to be the most effective, resulting in a remarkable esterification yield of 80% (higher than yield obtained in *n*-heptane). The enzymatic selective esterification of oleic acid with glycerol-based deep eutectic solvents acting as substrates and solvents (an example of reactive DES) was reported in 2015 by Zeng et al. [77] (Table 7, entry 2). Choline chloride or betaine can effectively change the reactivity of glycerol when they are mixed with a certain molar ratio. Results showed that, betaine has more moderate effects than ChCl on the lipase catalyst, and water content had an important influence on the esterification. The addition of ChCl to the system was found to improve the enzyme selectivity, thereby allowing the rapid and efficient synthesis of 1,3-diacylglycerol (DAG). Under these conditions, 43 mol% of 1,3-DAG was prepared over 1 h at 60 °C using ChCl/glycerol 1:2 as the solvent (Table 7, entry 2). In 2017 Guajardo et al. [78] reported a Novozym 435-catalyzed esterification between benzoic acid and glycerol performed in a ChCl/glycerol–water mixture, resulting in 99% conversion to *α*-monobenzoate glycerol product (Table 7, entry 3). The low viscosity of ChCl/glycerol–water mixtures was also exploited later for the continuous biocatalyzed esterification of glycerol with benzoic acid (Table 7, entry 3) [79]. These reactions were carried out in a fed-batch bioreactor at a substrate flow rate of 0.01 mL/min, with a maximum conversion of 90%. The fed-batch operation increased the conversion by 59% compared to the batch mode. After 10 days, only a 2% decrease was observed in the stability of the biocatalyst in continuous mode. In this second example the source of glycerol as substrate is the DES itself. Furthermore, to reinforce catalyst stability and to facilitate its recovery the immobilization of CALB in cross linked aggregates (CLEA) derivatives in Lentikats^®^ (lens-shaped polyvinyl alcohol hydrogel particles) was reported [80]. The CALB-CLEA immobilization leads to a robust biocatalyst (CLEA-CALB-LK) that remains very stable in low viscous non-conventional DES-buffer mixtures (Table 7, entry 3). This double immobilization can be successfully used in batch and continuous processes, reaching full conversion. Under these conditions, the derivatives display an improved stability (compared to the lipase-CLEA derivatives) and enable the reuse of the reaction media in continuous devices for at least six cycles under non-optimized conditions, enhancing the productivity. Then, in 2017, Wang et al. [81] reported an efficient synthesis of *n*-3 poly unsaturated fatty acids (PUFA)-enriched triacylglycerol (TAG) by the esterification of glycerol with *n*-3 PUFA in ChCl/urea 1:2 (Table 7, entry 4). There was a 1.2-fold increase of TAG yield in DES compared with that in the solvent-free system. Adsorption of the produced water by DES during esterification contributed to enhance the conversion efficiency by changing the reaction equilibrium. DES also served as an effective solvent for enriching the *n*-3 PUFA of TAG in the upper layer of reaction media. A TAG yield of 56% was achieved under optimal conditions, with an esterification degree of 93% with the following conditions: DES at 1:5 (mol/mol) of *n*-3 PUFA to glycerol, 3.30:1 (*v*/*v*) of DES to glycerol, 50 °C for 24 h.

In recent years, a series of investigations were conducted on the esterification of menthol with long-chain fatty acids. In 2018, Holtmann et al. began investigating DESs based on (-)-menthol and fatty acids (octanoic, decanoic and dodecanoic acid) as reaction media for the *Candida rugosa* (CRL) catalyzed self-esterification of the DES components to synthesize (-)-menthol fatty acid esters (Table 7, entry 5) [82]. The DES acts as reaction medium and substrate pool simultaneously without the need to add any solvent. The addition of water to the lipophilic DESs enhanced the reaction outcome, likely due to interfacial activation of the enzyme. In biphasic reaction systems with an addition of 10% (*w*/*w*) of water to the DES phase, the conversion (7 days, 35 °C) of octanoic, decanoic and dodecanoic acid reached 50%, 83% and 71%, respectively. Closer investigation of the reaction system revealed that water addition and stirring speed are interacting parameters. Furthermore, a product recovery strategy from the DES (−)-menthol/dodecanoic acid 3:1 was later demonstrated [101]. The product (−)-menthyl dodecanoate ester was separated from the DES reaction mixture and a second esterification reaction could be performed with the recovered (−)-menthol. Afterwards, the use of *rac*-menthol as part of the DES and the enantioselectivity of the enzyme was also studied by Paiva et al. [83]. *Rac*-menthol and lauric acid were the DES components and also the esterification reaction substrates. The DES reacted promptly when *Candida rugosa* lipase was added, and the conversions of *rac*-menthol reached up to 44% after 3 h of reaction and an enantiomeric excess of the product of 62% (Table 7, entry 5). Enzyme activity was dependent on the molar ratio of the substrates as well as on the water activity in the DES.

The applicability of hydrophobic (-)-menthol/decanoic acid 1:1 DES for glycolipid *i*CALB-catalyzed synthesis was implemented by Hollenbach et al. in 2020 [84]. Initial glucose monodecanoate yields, although very low (0.71%), were improved by using the above mentioned lipophilic-based DES (Table 7, entry 6). The polarity of the solvent was identified as crucial for glycolipid productivity. Furthermore, the reaction was also possible with free fatty acids instead of the thermodynamically preferred reaction with vinylated fatty acids. Moreover, the enzyme showed high stability and reusability in the hydrophobic DES without loss of activity for at least five reaction cycles.

Panthenol, also known as provitamin B5, is a bioactive molecule that has important applications in pharmaceutical and cosmetics industries [102]. The ability of panthenol/fatty acid mixtures to form eutectic mixtures, was successfully applied for the lipase-catalyzed selective synthesis of panthenyl monoacyl esters (PME) by direct esterification in a solvent-free approach [85]. The reaction yields (up to 84%) and PME selectivity (93–99%) makes this approach a useful way to prepare panthenyl monoacyl esters (Table 7, entry 7). Furthermore, the enzymatic activity of Novozym 435 in biocatalyzed process has been reported to remain unchanged after reuse of the catalyst over seven consecutive cycles. Enzymatic esterification of starch in DES was reported [86] using ChCl/EG 2:1 as reaction medium, Novozyme 435 as biocatalyst, and PEG 400 as phase transfer agent to synthesize starch decanoate, laurate, and palmitate. Results showed that the degrees of substitution (DS) were in the range of 0.07–0.19. The optimal esterification temperature for DS was 65 °C. The degradation of starch was found to be minor due to the use of long-chain fatty acid and relatively low reaction temperatures (Table 7, entry 8). The use of DESs in the lipase-catalyzed esterification of lactic acid with ethanol to afford ethyl lactate (an important bio-based solvent) was explored by Takaç et al. in 2019 [87]. ChCl/glycerol 1:2 was the most effective DES and provided 29% yield of ethyl lactate under the following conditions: 10% (*v*/*v*) of water content in DES, 3M of initial lactic acid concentration, 5M of initial ethanol concentration, 30 mg/mL of enzyme concentration (Novozym 435), 50 °C and 200 rpm agitation rate in 72 h (Table 7, entry 9). Individual and combined effects of the reaction medium components on the enzyme activity were investigated and it was discovered that DES enhances enzyme activity while reactants inhibited it.

### 5.2. Transesterifications Reactions with Isolated Enzymes

The first example for this class of reactions was reported by Kazlauskas et al. in 2008 [88], where they tested the lipase-catalyzed transesterification of ethyl valerate with 1-butanol using different forms of CALB in eight DES. Among these runs, the transesterification reaction in EAC (*N*,*N*-diethylethanolammonium chloride)/Gly 3:2 with *i*CALB (Novozym 435, 60 °C for 24 h) showed 93% conversion and less than 0.5% glyceryl ester formation (Table 8, entry 1). The initial specific activity of *i*CALB in EAC/Gly 3:2 after 15 min of reaction was of 50 (135%), compared to 37 (100%) for toluene. They also later reported [89] another *i*CALB-catalyzed transesterification of ethyl valerate, this one with 2-butanol. The initial activity of *i*CALB in the transesterification of ethyl valerate with 2-butanol in DESs was comparable to that in toluene. Typical conversions were 10–45%, but the best solvent ChCl/urea 1:2 gave a conversion of 74% (enantioselectivity, *E* = 3.6). Immobilized CALB was most active in ChCl/urea 1:2 in contrast to toluene and ionic liquid BMIM[TF_2_N]. The enantioselectivity of *i*CALB toward 2-butanol is poor (*E* = 9.9 in toluene at 40 °C), likely due to the difficulty in distinguishing the methyl and ethyl substituents at the stereocenter (Table 8, entry 2). This enantioselectivity decreased by a factor to two or more in both BMIM[Tf_2_N] and in the DESs used. The advantages and limitations of several DESs as ‘green solvents’ for transesterifications using *i*CALB (Novozym 435) as catalyst were analyzed by Villeneuve et al. [90]. The transesterification of vinyl laurate was chosen as a model reaction and the influence of substrate polarity was assessed using alcohols of various chain lengths (Table 8, entry 3). Results showed that grinding of immobilized lipase was an essential parameter for good lipase activity. Moreover, some hydrogen-bond donor components from the DES can compete with the alcoholysis reaction. Side reactions were observed with DES based on dicarboxylic acid or ethylene glycol, leading to some limitations in their use. However, the results showed that other DESs such as ChCl/urea 1:2 and ChCl/glycerol 1:2 could exhibit high activity and selectivity. The best DES’s specific activity and stability, up to five days incubation time, were analyzed and compared with conventional organic solvents. Experiments revealed that *i*CALB is less influenced by the chain length of alcohol in DES than organic solvents and it preserves its activity with minimally destructive to protein structure. A series of *i*CALB-catalyzed transesterifications between different phenolic esters as substrates of different polarities in DESs was reported by Durand et al. [91]. They showed that water content could dramatically improve the lipase activity and change the reactivity of phenolic substrates. Indeed, very low conversions (<2%) were observed in pure DES, whereas in DES–water binary mixtures, quantitative conversions were achieved. After investigating the role of various parameters, such as the substrate concentration and ratio, pH or thermodynamic activity of water, the effect of the presence of water in pure DES based on urea or glycerol was discussed. A multigram scale experiment in ChCl/urea 1:2 with 10% water content was conducted to synthesize the lipophilized phenolic compound in high yield. The reaction was realized in the presence of 3 g of *p*-coumarate (40 mM) with 1-octanol in a 1:6 molar ratio, using 10 mg/mL of biocatalyst. Under these conditions, 97% of the substrate was converted after 72 h of reaction, with a yield of 93%, which confirmed the viability of this synthesis for a large-scale application (Table 8, entry 4). Later [103], they examined the role of the different components involved in these mixtures (ChCl, urea, and water) in the medium’s functional properties, thus optimizing the process. By varying the urea concentration, it has been shown that urea has a denaturing effect on the enzyme and a positive effect on the selectivity of the reaction. Preparation of methyl gallate, which is an important phenolic acid ester for pharmaceutical industry, was carried out by Novozym 435-catalysed transesterification of propyl gallate (PG) with methanol in DES by Takaç et al. [92]. Reaction parameters governing substrate molar ratio, enzyme concentration, temperature and agitation rate were investigated batch-wise in ChCl/glycerol–water binary mixtures. The results were evaluated in terms of conversion of propyl gallate, yield of methyl gallate and hydrolysis of the substrate to gallic acid. A total of 10% (*w*/*w*) of water was found to be favorable in the reaction medium. The highest yield was 60% after 120 h of transesterification at propyl gallate/methanol molar ratio of 1:6, enzyme concentration of 40 g/L, 50 °C and 200 rpm (Table 8, entry 5). Not only DES–water binary mixtures but also DES−DMSO cosolvent system have been successfully used as reaction medium in the case of enzymatic acylation of dihydromyricetin (DMY) catalyzed by the immobilized lipase from *Aspergillus niger* (ANL@PD-MNPs) by Lou et al. [93]. The cosolvent mixture, (ChCl/Glycerol 1:2)-DMSO (1/3 *v*/*v*) proved to be the optimal medium. With the newly developed cosolvent, the enzymatic acylation of DMY achieved a reaction rate of 11.1 mM/h and a conversion of 91%. The supported enzyme catalyst is stable and recyclable in this cosolvent, offering 90% conversion rate after five times (Table 8, entry 6). Albeit having used a non-optimized process, Siebenhaller et al. [94] reported a first step for the production of sustainable glycolipids through Novozym 435-catalyzed transesterification of glucose with fatty acids vinyl esters in ChCl/hydrolysate. The sugars were extracted from beechwood lignocellulose, a renewable and frequently available resource. By the formation of a eutectic mixture, the sugars contained are made easily available in the reaction solvent. This is an elegant way to avoid the negative effect of the low solubility of sugars in other water-free solvents. The maximum yield obtained under optimized conditions was of 4.81%, but there is great potential to further optimize the system (Table 8, entry 7). They also demonstrated that honey or agave syrup can be used simultaneously as solvents and substrates for the enzymatic sugar ester production. Thus, an enzymatic transesterification of four fatty acid vinyl esters was accomplished in ordinary honey and agave syrup with *i*CALB. The proposed process is not optimized, but it demonstrates that the reaction system is very versatile and stable [104]. Then, to further obtain glycolipids Siebenhaller et al. [105] used beech wood cellulose fiber hydrolysate as a sugar component, as part of the DES reaction system and as a carbon source for the microbial production of the fatty acid component. These fatty acids were gained from single cell oil produced by the oleaginous yeast *Cryptococcus curvatus*, cultivated with cellulose fiber hydrolysate as a carbon source. Then, immobilized *Candida antarctica* lipase B was used as the biocatalyst in DES to perform a transesterification of fatty acids methyl esters with sugars. Using this approach, sugar esters were successfully synthesized and were 100% based on lignocellulosic biomass. 1-Caffeoylglycerol (1-CG) is a high soluble hydrophilic ester, with a strong activity to prevent skin cancer caused by ultraviolet. An enzymatic synthesis in ChCl/urea 1:2 in a microreactor was reported by Wang et al. [95] under continuous microflow conditions. A maximum 1-CG yield of 96% was obtained under the optimized conditions: temperature of 65 °C, flow rate of 2 μL/min, methyl caffeate concentration of 50 g/L. Compared to the batch reactor (91%, 10 h), the reaction time was shortened by a quarter, with a 5% increase of yield. In these conditions, the lipase (Novozym 435) can be reused 20 times (Table 8, entry 8). A biocatalyzed transesterification in ChCl/urea 1:2 of vinyl butyrate with tyrosol was reported by Stamatis et al. [96], making use of a novel support for lipase B (from *Pseudozyma antarctica*) catalyst based on hybrid nanoflowers (copper (II) or manganese (II) ions). The effect of the addition of carbon-based nanomaterials, namely graphene oxide and carbon nanotubes, as well as magnetic nanoparticles such as maghemite, on the structure, catalytic activity, and operational stability of the hybrid nanobiocatalysts (CALB/HNFs) were also reported. In all cases, the addition of nanomaterials during the preparation of hybrid nanoflowers increased the catalytic activity and the operational stability of the immobilized biocatalyst (Table 8, entry 9). In 2013, Zhao et al. [97] introduced choline-based DES as lipase-compatible solvents for the enzymatic preparation of biodiesel from soybean oil through transesterification with methanol. They evaluated different eutectic solvents and different lipases, as well as the study of reaction parameters (i.e., methanol concentration, Novozym 435 loading and reaction time). Up to 88% triglyceride conversions in 24 h was achieved at 50 °C. The enzyme could also be reused for at least four times without losing significant activity (Table 8, entry 10). Afterwards, Hao et al. [98] reported an enzymatic transesterification from the waste oil and ethanol to produce fatty acid ethyl esters (FAEE). The process was investigated under three different conditions: ultrasound-assisted, in DES and ultrasound-assisted in DES. The experimental results showed that the conversion of FAEE was significantly improved, and the reaction time shortened (from 24 h to 1.5 h) under the ultrasound-assisted DES system, the ultrasonic power and reaction temperature had synergistic effect on the conversion of FAEE. The conversion rate reached 94% with the optimized reaction conditions (59 °C, ultrasonic power 80 W, 137 min with addition of 6.4% lipase and 4 mmol alcohol for 1 g of oil, Table 8, entry 11). Finally, in 2021, Delavault et al. [99] performed a comparison between 16 commercially available lipase formulations in a DES made of sorbitol and choline chloride (1:1) and investigated the factors affecting the conversion with the aim of optimizing the transesterification reaction between D-sorbitol and vinyl laurate. Thus, using 50 g/L of Novozym 435 at 50 °C, the optimized synthesis of sorbitol laurate allowed to achieve 28% molar conversion of 0.5 M of vinyl laurate to its sugar alcohol monoester when the DES contained 5% *w*/*w* water (Table 8, entry 12). After 48 h, the glycolipid product was separated from the media by liquid–liquid extraction, purified by flash-chromatography and characterized. They also provided initial proof of scalability for this process, using a 2.5 L stirred tank reactor for a batch production reaching 25 g/L in a highly viscous two-phase system.

## 6. Miscellaneous Enzymatic Reactions

Throughout the years, enzymatic synthesis with the use of DESs as (co)solvents has been extended to a number of reactions which do not fall into any of the categories presented in the previous sections. Successful examples have been reported in the field of amine and amide chemistry. The formation of the amide bond is relevant to medicinal chemistry, because it is a widespread motif in small molecules used as active pharmaceutical ingredients (APIs), and of course because of its application to the synthesis of oligopeptides. The conversion of fish oil fatty acid esters ethyl docosahexaenoate and eicosapentaenoate into the corresponding ethanolamides (Table 9, entry 1) falls into the first of the two purposes, since these polyunsaturated long alkyl chain amides have been reported to play several biological functions. For this reaction, the activity of immobilized *Candida antartica* lipase B Novozym 435 in ChCl-based DESs was studied by taking into account the effect of different parameters, such as the nature of the eutectic mixture, the water content, the reaction temperature and time, the molar ratio between the fish oil esters mixture and ethanolamine [106]. Best results in terms of conversion were obtained with ChCl/Glu 5:2 with a H_2_O content of 8.50% *w*/*w*; other DESs afforded lower yields. The study represented an improvement over previous methods, not only in terms of replacement of VOCs and increasing of the reaction yields within short reaction times at moderate temperatures, but also because the product was recovered and separated from the DES-ethanolamine mixture by simple centrifugation, thus avoiding the need for any further purification, paving the way to the possible application on an industrial scale [106].

The amide bond formation for the synthesis of bioactive compounds was also exploited in the synthesis of API cefaclor (Table 9, entry 2) by reaction of D-phenylglycine methyl ester and 7-amino-3-deacetoxycephalosporanic acid, catalysed by enzyme penicillin acrilase immobilized on magnetic nanocrystalline cellulose [107]. Among the DESs screened, in combination with phosphate buffer as cosolvent, ChCl/EG 1:2 gave the best results. The investigation on the reaction parameters was oriented to the optimization of the yield of cefaclor and to the suppression of the undesired hydrolysis of the methyl ester starting material to the corresponding carboxylic acid. In this regard, the use of DES allowed the issue of the low solubility of the amine reaction partner in pure phosphate buffer to be addressed, previously reported by the same authors [108], resulting in an improved efficiency of the synthesis of cefaclor towards the undesired hydrolysis [107]. The same group also supported a different enzyme, papaine, onto the same magnetic nanocrystalline cellulose support, to be applied in peptide synthesis. Specifically, the dipeptide *N*-(benzyloxycarbonyl)-alanyl-glutamine (Z-Ala-Gln) was synthesized by the condensation of *N*-(benzyloxycarbonyl)-alanyl methyl ester (Z-Ala-OMe) with glutamine (Table 9, entry 3), using ChCl/U 1:2 with 10% *v/v* H_2_O content [109]. The combination of the use of the DES as reaction medium, in combination with the immobilization of papain onto the support, allowed an increase in the stability of the enzyme itself, resulting in high yields of Z-Ala-Gln and the possibility to recycle the catalytic system up to five runs without significant loss in activity [109]. Similar results were later obtained in the analogous synthesis of parent dipeptide *N*-(benzyloxycarbonyl)-alanyl-histidine (Z-Ala-His, Table 9, entry 4), with ChCl/U 1:2 (17% *v*/*v* H_2_O content) and phosphate buffer as cosolvent. In this case, it was highlighted that the catalytic system could be easily recovered at the end of the reaction by taking advantage of its magnetic properties and reused up to seven runs [110]. Finally, the synthesis of Z-Ala-Gln was achieved also in ChCl/Gly 1:2 (with 20% *w*/*w* H_2_O content) and ethanol as cosolvent, by using a manganese-driven papain-surfactant nanocomposite. The authors also performed a techno-economical and environmental analysis of a possible implementation of the methodology on a production scale, with operating volumes of the reactor in the range 10^2^–10^4^ L [111].

For what concerns the synthesis of amines, amine transaminases were successfully employed in a DES for the first time in 2019. The methodology consisted in a tandem Suzuki coupling–transamination reaction, conducted in a DES-phosphate buffer medium, which was fitting the requirements of both the palladium-catalyzed and the biocatalytic step (Table 9, entry 5). Remarkably, the one-pot procedure afforded enantiopure biaryl amines (*ee* values as high as 99%) as the final products, without the need for extensive purification, as a simple solvent extraction was sufficient to separate the hydrophilic components (including those of the DES) from the product, which then just required a filtration over silica [112]. Some efforts to the application of transaminase biochemistry in DESs have been dedicated by the group of *He* to the conversion of furfural and 5-hydroxymethylfurfural (Table 9, entry 6) into the corresponding furfuryl amines, in the framework of the valorisation of lignocellulosic biomass, such as sugarcane bagasse [113], corncob [114], and sugar [115]. The biomass was first processed to obtain the furan derivatives, either through a tin-based heterogeneous catalyst, obtained exploiting waste shrimp shells, in ChCl/EG or ChCl/Gly DESs and H_2_O as cosolvent [113,114] or through dehydration in a malic acid (MA)Gly/betaine (Bet) DES with H_2_O as cosolvent, in the absence of any catalyst [115]. Then, in the same reaction medium, recombinant *E. coli* whole cells were employed to promote the transamination of the aldehyde into amine functionality (Table 9, entry 6). The reaction parameters were extensively explored and both the catalyst and the reaction medium were successfully recycled. These works are strongly oriented towards a process scale application of chemoenzymatic synthesis in DESs, in a circular economy context.

On the other hand, the manipulation of biomass-derived compounds has also been investigated in terms of *N*-methylation of the polysaccharide chitosan (Table 9, entry 7). One of the major drawbacks of this pharmaceutically relevant reaction is its low selectivity, which leads to mixtures of *O*- and *N*-methylated products; additionally, a frequent side reaction is the scission of the chitosan polymer, due to the strong conditions required, lowering the molecular weight of the final product. Two consecutive studies disclosed the possibility to employ milder reagents and drastically lower the temperature of the process, by switching from a chemical synthesis, using iodomethane as methylating agent and NaOH as base, to a biocatalytic one, in DES/water mixtures [116,117]. The family of enzymes selected for the transformation was lipase (from *Burkholderia cepacian* or *Candida rugosa*), which displayed a good activity in converting chitosan into the desired methylated products, using dimethyl carbonate as green methylating agent and in the absence of base additive. Again, the choice of the eutectic mixture was able to affect the reactivity, as with binary DES ChCl/U 1:2 *N,N*-dimethylation was mainly observed, alongside with some *O*-methylation [116], while in ternary ChCl/U/Gly 1:2:1 a high degree of the desired *N,N,N*-trimethylation was achieved [117].

The valorisation of lignocellulosic biomass is involved also in the decarboxylation of phenolic acids, such as *p*-hydroxycinnamic acids (Table 9, entry 8), which can be obtained from the hemicellulose fraction of plants [118]. The biocatalytic conversion of the α,β-unsaturated acids into *p*-hydroxy styrenes, with phenolic acid decarboxylase enzymes, is affected by the low solubility of the substrates in water, on one side, and by the inhibition of the enzyme activity in conventional organic solvents, on the other side. In this framework, the application of ChCl/U 1:2 DES as cosolvent, in combination with water, enabled the efficient synthesis of the desired styrenes, under mild conditions and with high substrate loadings [119]. The synthetic relevance of the methodology was later underlined by the successful implementation of tandem transformations, in which the biocatalytic step was followed by a Pd-catalyzed Heck cross-coupling or a Ru-catalyzed olefin metathesis, in the same water–DES reaction environment [120,121].

Biocatalysis in DESs has been exploited also in the field of phospholipids, specifically for the transphosphatidylation of phosphatidylcholines (Table 9, entry 9). This reaction involves the substitution of the choline polar head of the substrate. In a first study, the target product was phosphatidylserine, which was obtained using enzyme phospholipase D in ChCl/EG 1:2 as the reaction medium, after extensive screening of possible ChCl-based eutectic mixtures [122]. The use of the DES with a minimal water content (5% *w*/*w*) allowed circumvention of hydrolysis of the phosphatidylcholine starting material to the corresponding acid, which is a reaction catalyzed by phospholipase D as well, without resorting to VOCs. Furthermore, it was possible to recycle the enzyme up to 10 runs without significant loss in activity [122]. Remarkably, a more recent study applied the same methodology, but employing the DES not only as reaction medium, but as reagent, achieving the substitution of the choline polar head of the substrate with the HBD component of the eutectic mixture (Table 9, entry 10). Thus, phosphatidylethylene glycol and phosphatidylglycerol were successfully synthesized and recovered by simple filtration, as they resulted to be insoluble in the reaction mixture [123].

Finally, biocatalysis in DESs has been applied to the carboligation of aldehyde substrates (Table 9, entry 11): benzaldehyde lyase was employed to perform a homocoupling of alkyl and aryl aldehydes into α-hydroxyketones [124]. Satisfactory yields were obtained with ChCl/Gly 1:2, with 40% *v*/*v* phosphate buffer as cosolvent: it was indeed shown that lower amounts of aqueous medium led to the inactivation of the enzyme. The enantioselectivity of the reaction was comparable to the same reaction performed with VOCs instead of DESs [124]. It should also be reported that another reaction involving carbonyl groups, the aldol reaction, was investigated in its enzymatic version in some DESs, in the framework of a study which was, however, rather focused on the application of imidazolium-based ionic liquids [125].

**Table 9 molecules-28-00516-t009:** Miscellaneous enzymatic reactions.

Entry	Miscellaneous Enzymatic Reactions	Ref.
1	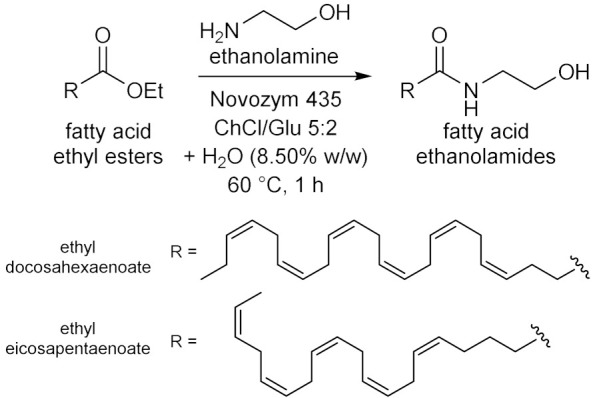	[106]
2	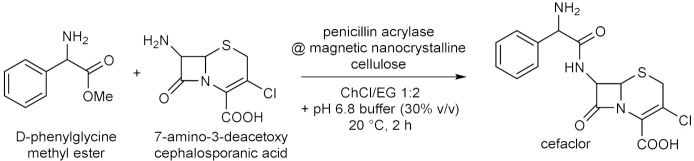	[107]
3	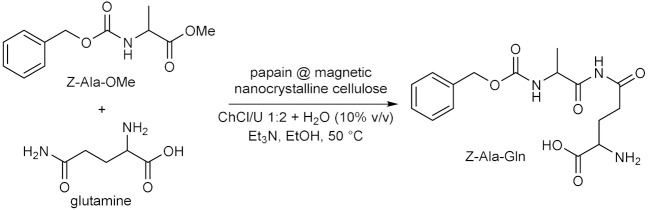	[109]
4	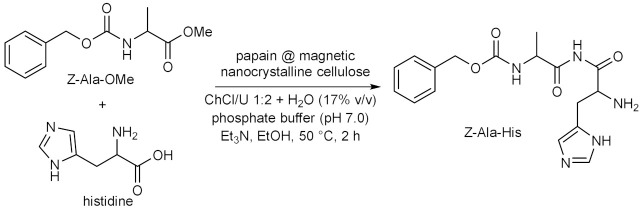	[110]
5	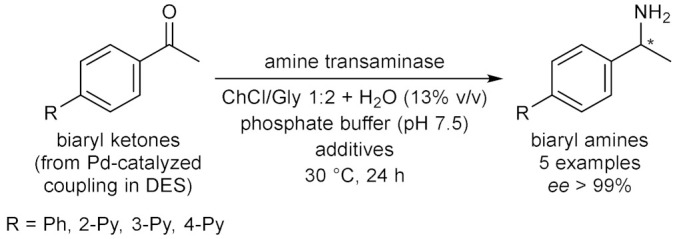	[112]
6	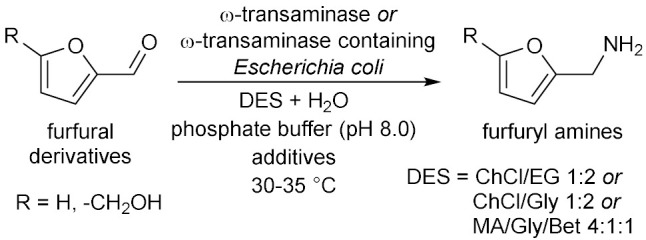	[113,114,115]
7	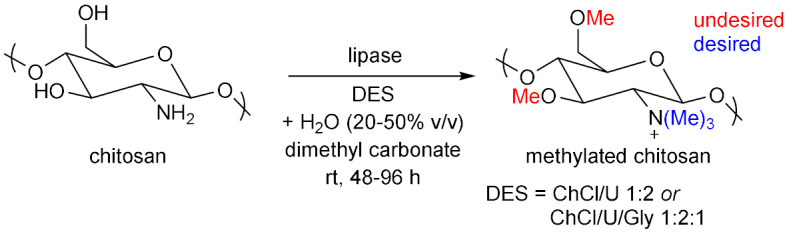	[116,117]
8	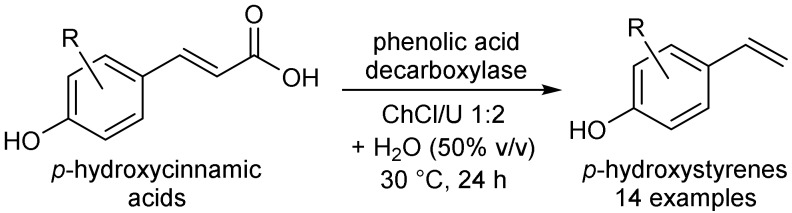	[119]
9	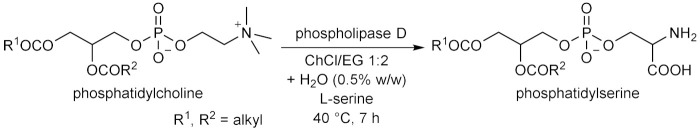	[122]
10	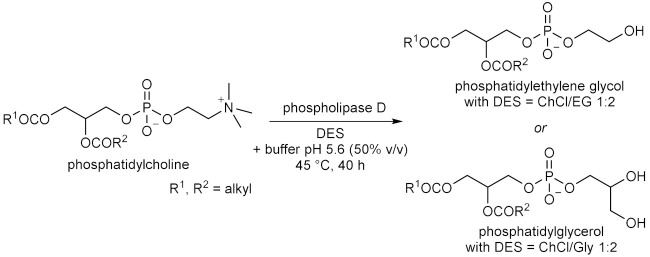	[123]
11	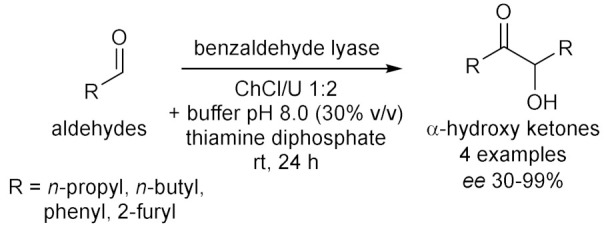	[124]

## 7. Conclusions

The use of non-conventional solvents in biocatalytic transformations represents a remarkable advance to develop more sustainable processes and to fully exploit the potential of enzymes either native isolated or engineered. Among the neoteric solvents, DES and NADES are well placed to overcome some of the main drawbacks related to the use of organic solvents or, more recently, to ILs. DES are simple and cost-effective to produce, show low volatility and have been proven to be more biocompatible [126]. In this review, we reported and discussed the main beneficial effects of using DES as co-solvent or even as only solvent. Redox, hydrolysis, esterification, transesterification reactions and various additional transformations catalyzed either by isolate enzymes or by whole cells in the presence of DES were shown to provide high conversions and shorter reaction times depending on the nature and percentage of DES in comparison with reactions run in water or phosphate buffer. Enantioselectivity is often excellent; sometimes stereoinversion is observed in DES. In whole cells catalysis, DES are supposed to increase cell permeability and thus favor the catalytic activity of enzymes. In the future, further investigations would be necessary to deeply understand the effect of DES on the conformational changes and on the activity of the enzymes. Looking at the possible application of biocatalysis in DES at a production scale, such as in the pharmaceutical industry, in the valorization of biomass or in the production of fine and specialty chemicals, a major hurdle could be represented by the high viscosity values exhibited by the eutectic mixtures, which could make it very hard to process the solvent–enzyme–substrate mixtures in a production plant. However, a possible solution to this problem may already be at hand, for most of the methodologies covered in this review: indeed, the addition of even a small amount of water to the DES is responsible for a significative decrease in the viscosity of the system [127,128]. The actual viscosity and ease of manipulation should then not be overlooked, when studying a biocatalytic synthetic methodology in DES with potential applications at an industrial scale [129]. To conclude, NADES are very promising solvent media for biocatalysis; nevertheless, current efforts need to be addressed to the investigation of the selectivity displayed by enzymes, to gain deeper insights into mechanicist aspects at the molecular level. If we consider all the other environmental benefits of applying DESs in the role of solvents, such as low vapor pressure (reduced air pollution), nonflammability (process safety), and nontoxicity, DESs represent excellent candidates for use in environmentally-friendly biocatalysis.

## Figures and Tables

**Table 1 molecules-28-00516-t001:** Reductions with isolated enzymes.

Entry	Reductions with Isolated Enzymes	Ref.
1	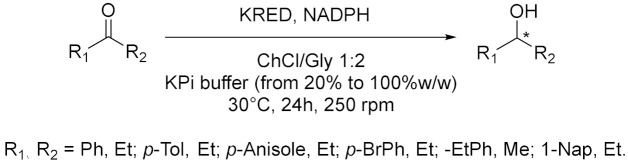	[12]
2	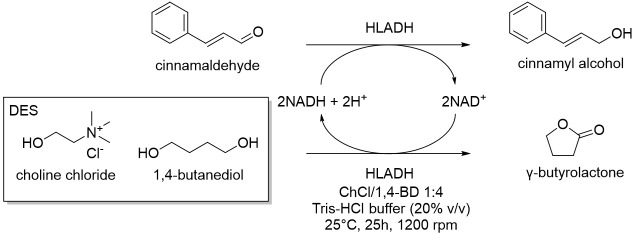	[13]
3	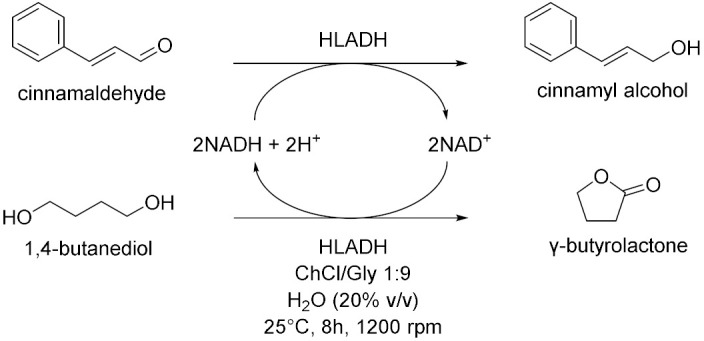	[14]
4	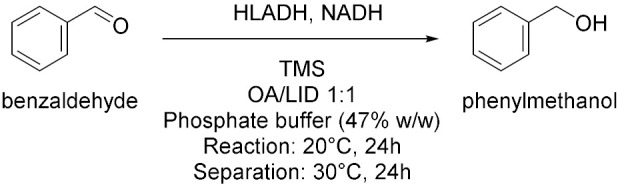	[15]

**Table 2 molecules-28-00516-t002:** Whole-cells catalysed reductions.

Entry	Whole-Cells Catalysed Reductions	Ref.
1	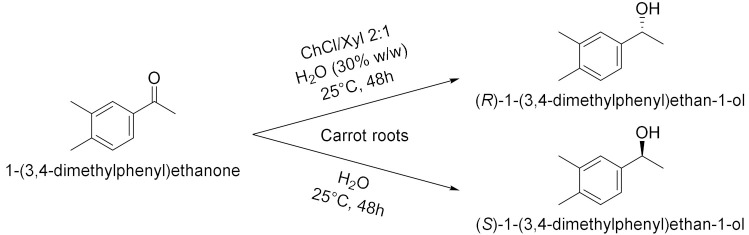	[16]
2	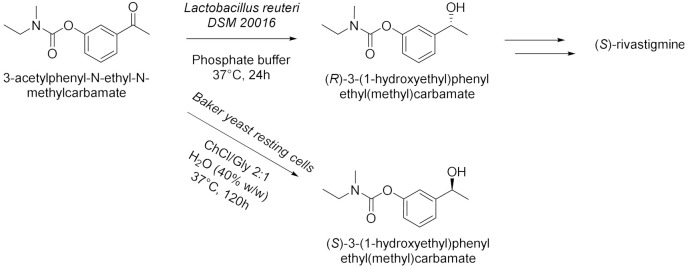	[17]
3	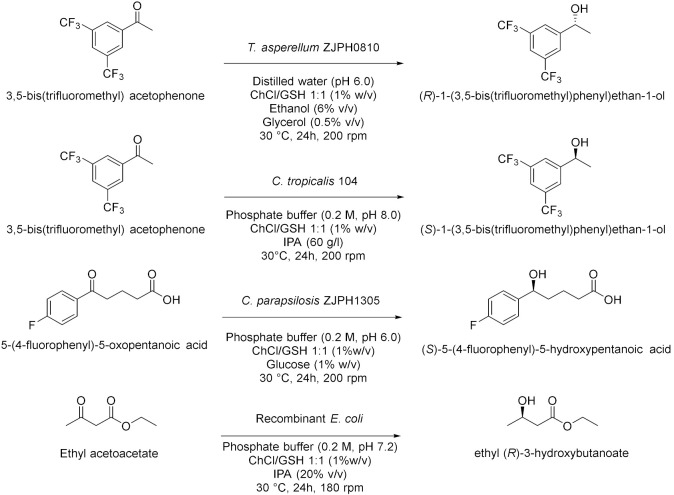	[18]
4	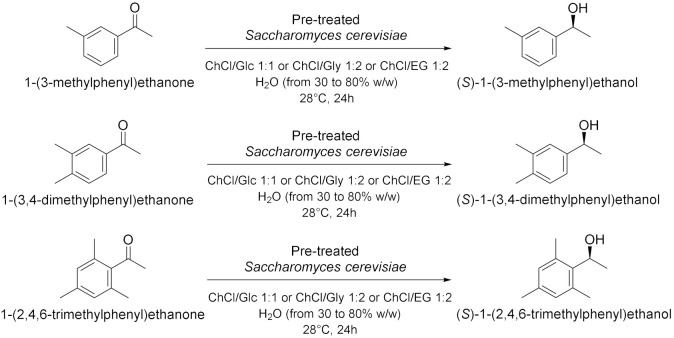	[19]
5	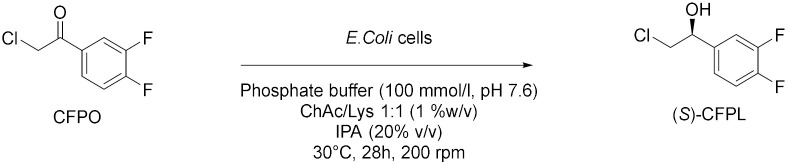	[20]
6	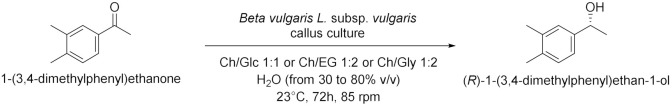	[21]
7	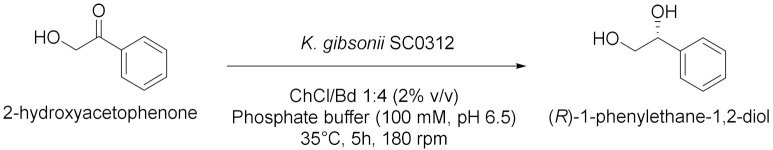	[22]
8	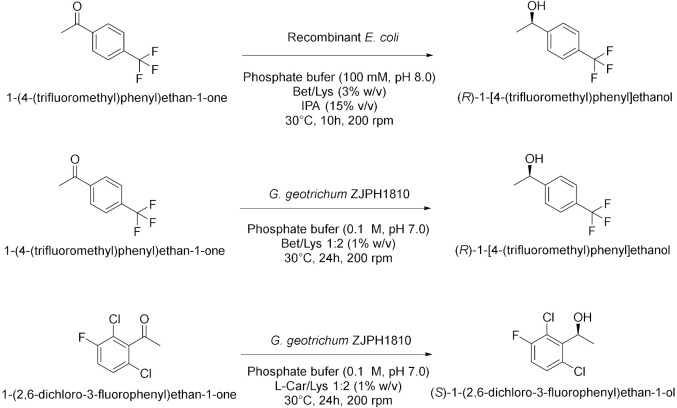	[23]
9	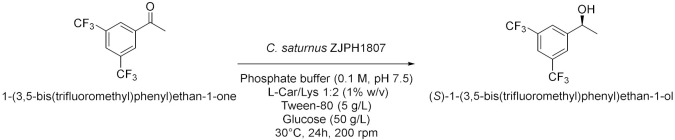	[24]
10	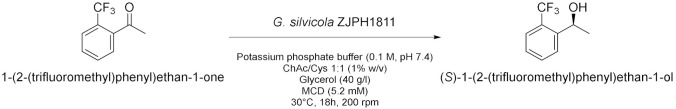	[25]
11	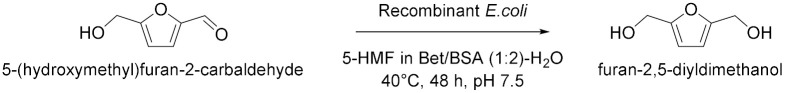	[26]

**Table 5 molecules-28-00516-t005:** Hydrolyses with isolated enzymes.

Entry	Hydrolyses with Isolated Enzymes	Ref.
1	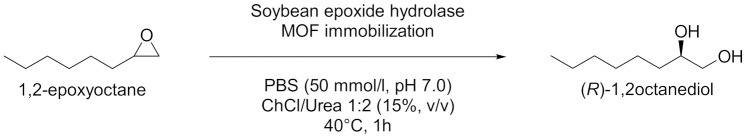	[56]
2	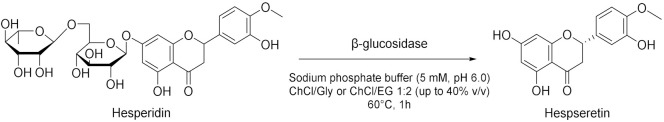	[57]
3	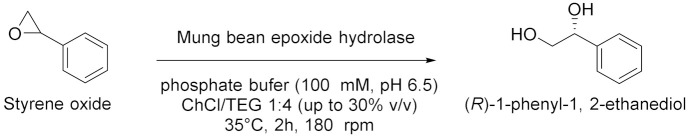	[58]
4	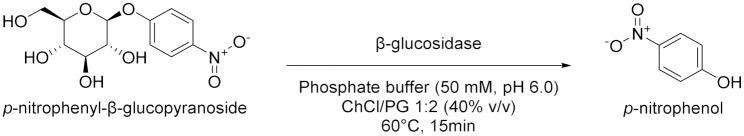	[59]
5	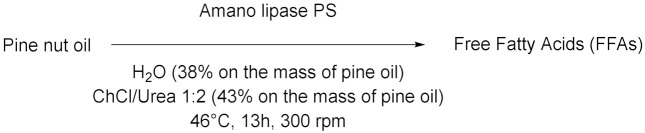	[60]
6	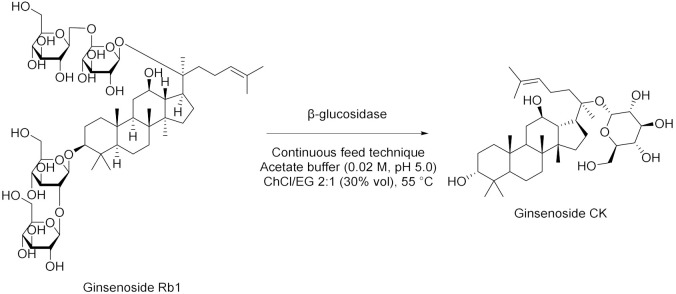	[61]
7	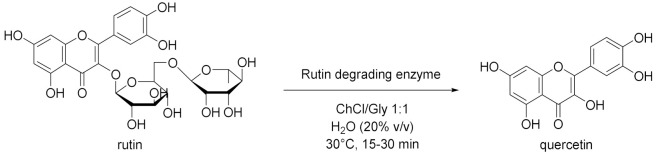	[62]

**Table 6 molecules-28-00516-t006:** Whole-cells catalysed hydrolyses.

Entry	Whole-Cells Catalysed Hydrolyses	Ref.
1	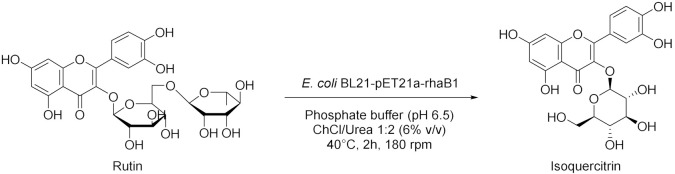	[63]
2	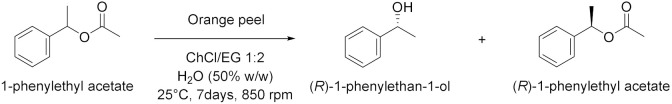	[64]

**Table 7 molecules-28-00516-t007:** Esterification reactions with isolated enzymes.

Entry	Esterification Reactions with Isolated Enzymes	References
1	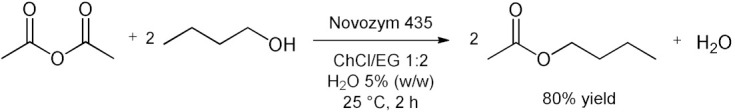	[76]
2	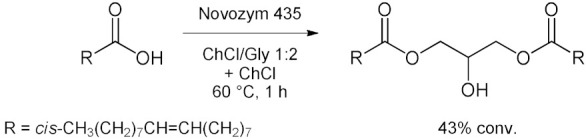	[77]
3	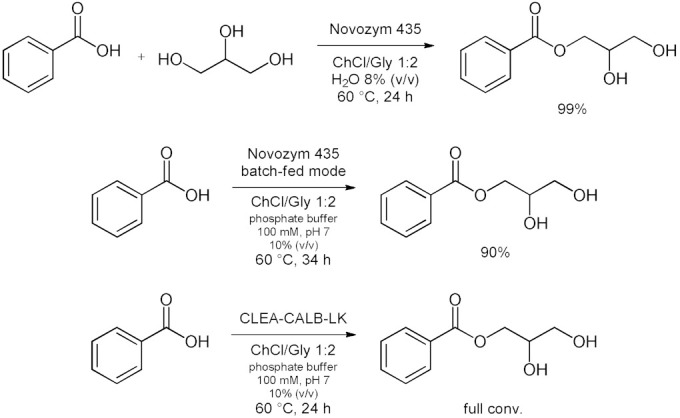	[78][79][80]
4	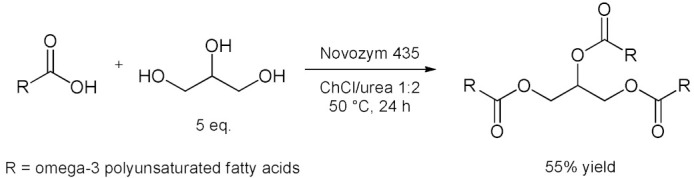	[81]
5	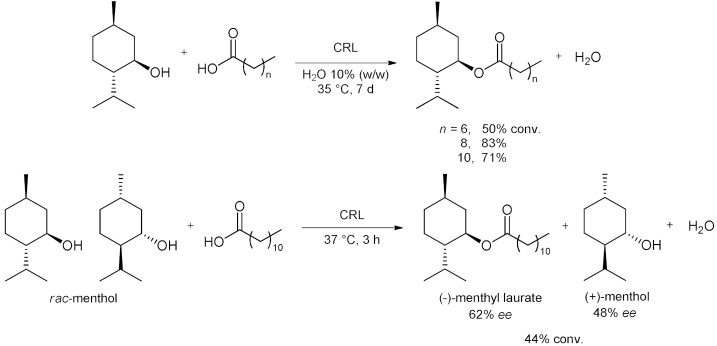	[82][83]
6	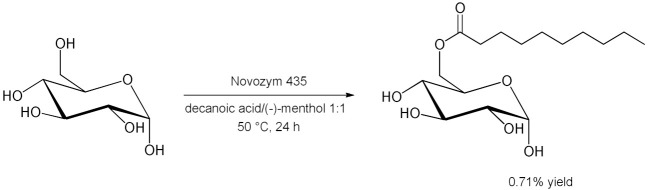	[84]
7	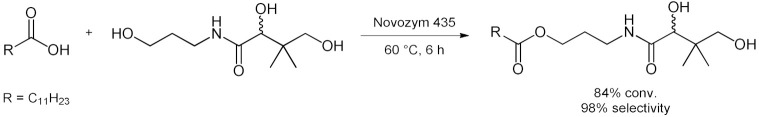	[85]
8	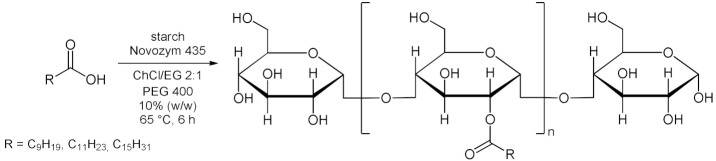	[86]
9	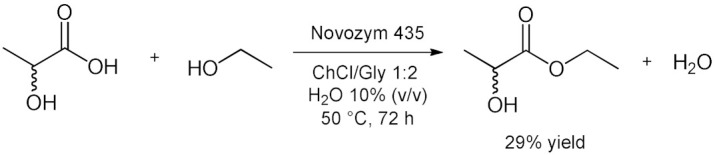	[87]

**Table 8 molecules-28-00516-t008:** Transesterifications reactions with isolated enzymes.

Entry	Transesterifications Reactions with Isolated Enzymes	References
1	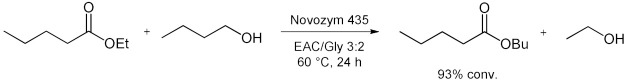	[88]
2	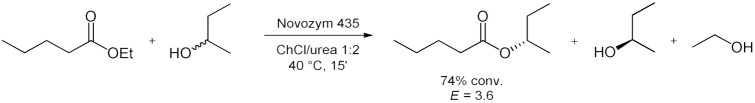	[89]
3	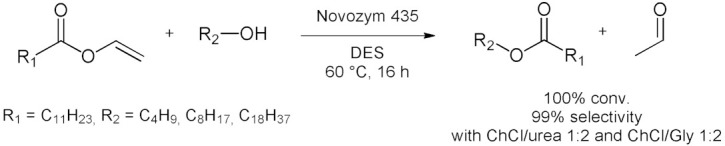	[90]
4	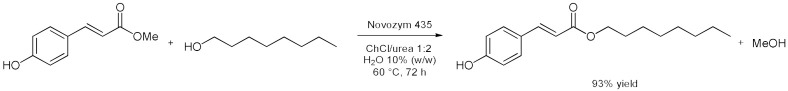	[91]
5	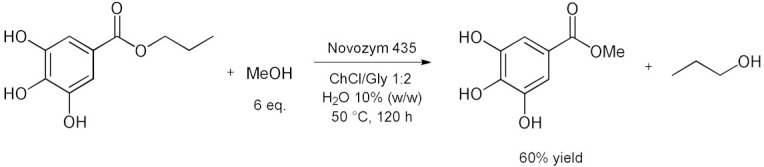	[92]
6	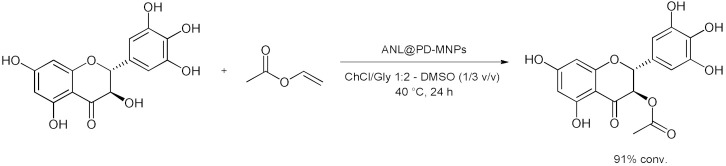	[93]
7	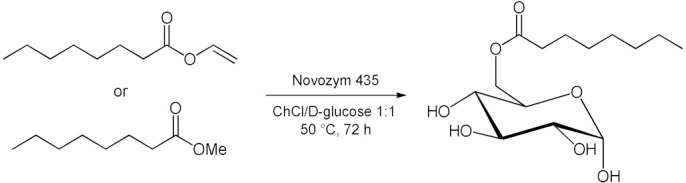	[94]
8	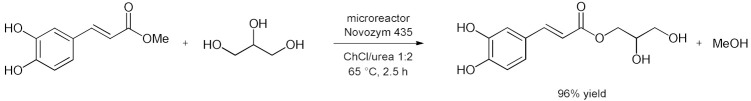	[95]
9	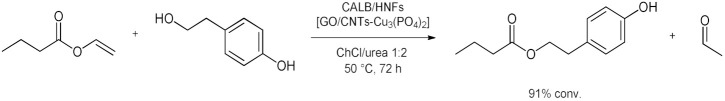	[96]
10	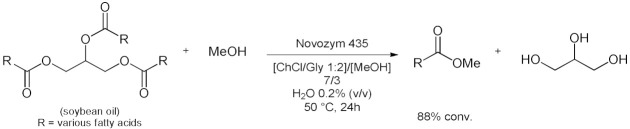	[97]
11	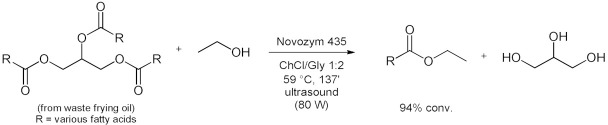	[98]
12	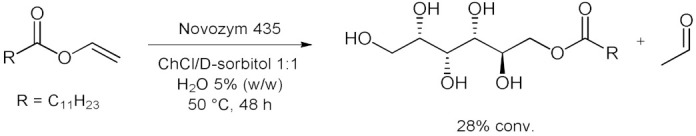	[99]

## Data Availability

Data sharing not applicable. No new data were created or analyzed in this study. Data sharing is not applicable to this article.

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
