# Peer review of "Combination of Enzymes and Deep Eutectic Solvents as Powerful Toolbox for Organic Synthesis"

_molecules, 2023, doi:10.3390/molecules28020516_

Round 1

Reviewer 1 Report

The authors wrote a comprehensive review of the combination of enzymes and deep eutectic solvents in organic synthesis. This work is interesting due to the complete and updated information. However, some aspects need to be clarified:

1.      There are many reviews about DES. How does this work differ from those already published?

2.      On page 2, lines 95-97, the authors note that the investigators used a DES whose ChCl/glycerol ratio was 1:9. In this case, considering the excessive amount of glycerol, wouldn't it be convenient to work only with glycerol as a solvent?

3.      On page 3 lines 103-1010, the authors discuss the Thermomorphic Multiphasic System. Please explain. What is the advantage of these DES?

4.      On page 4 the authors discuss modifying chirality using DES. Please write a short introduction on chiral biotransformations.

5.      Why do DES modify chirality in enzymatic biotransformations? What is your function? Are there chiral DES?

6.      On page 4 lines 132-133. The authors mention based-DES/water, but do not indicate which are the acceptors and donors of hydrogen.

7.      On page 4, lines 146-148, explain. What is the effect of water on chirality?

8.      On page 14, lines 441-445, the authors point out that DES have a beneficial effect on whole cells. Why does this behavior happen?

9.      On page 20, line 627, the correct word is "fed-batch"

10.  Please check the references. For example, in reference 37 the journal is not mentioned.

Author Response

Detailed responses to the reviewers’ and editor comments are listed here below.

Reviewer n.1

The authors wrote a comprehensive review of the combination of enzymes and deep eutectic solvents in organic synthesis. This work is interesting due to the complete and updated information. However, some aspects need to be clarified:

  1. There are many reviews about DES. How does this work differ from those already published?

As mentioned in the introduction, this review intends to give an overview on the many biosynthetic transformations realized in or with DES and includes not only redox reactions but also transesterifications, hydrolysis and all those conversions not included in the previous classes but worth of mention for their innovative potential. Additionally, the layout has been chosen with the specific purpose to focus on an IFG (Interconversion of Functional Groups) approach with the aim of providing organic chemists with a toolbox of new and greener alternatives to traditional ones. So, from one side the review updates the increasing number of references in the literature on this new emerging topic, and from the other it offers a toolbox for organic synthesis methodologies.

  1. On page 2, lines 95-97, the authors note that the investigators used a DES whose ChCl/glycerol ratio was 1:9. In this case, considering the excessive amount of glycerol, wouldn't it be convenient to work only with glycerol as a solvent?

We agree with the referee, however we are simply reporting what the authors stated in their paper.

  1. On page 3 lines 103-1010, the authors discuss the Thermomorphic Multiphasic System. Please explain. What is the advantage of these DES?

We added a specification in the text to better explain the advantage of Thermomorphic Multifasic Systems. Specifically, the shift between mono and biphasic systems allows for a better recovery of the product and of the enzyme.

  1. On page 4 the authors discuss modifying chirality using DES. Please write a short introduction on chiral biotransformations.

A short introduction on chiral biotransformation has been introduced at the end of the introduction.

  1. Why do DES modify chirality in enzymatic biotransformations? What is your function? Are there chiral DES?

Many chiral DES are reported in the literature, however the effect on enantioselectivity is not strictly linked to the chirality of the DES itself but on the effect DES have on the stability of various isoforms of the enzyme. As reported on pg. 4, sometimes in DES the activity of the enzymes leading to one enantiomer is completely suppressed, thus affording the opposite enantiomer of what is obtained in PBS.

  1. On page 4 lines 132-133. The authors mention based-DES/water, but do not indicate which are the acceptors and donors of hydrogen.

We mentioned choline-chloride based DES (the hydrogen bond donor may be different). When water is added to a DES, it can either be an acceptor or donor of hydrogen bonds. And the network of hydrogen bonds becomes even more complex.

  1. On page 4, lines 146-148, explain. What is the effect of water on chirality?

There is not a direct effect of water on chirality, however the presence of an increasing amount of water can loosen the tight network of hydrogen bonds and affect the activity of the various isoforms of the enzyme, with an indirect effect on chirality.

  1. On page 14, lines 441-445, the authors point out that DES have a beneficial effect on whole cells. Why does this behavior happen?

In principle, when the compounds that constitute the DESs are primary metabolites, namely amino acids, organic acids, sugars, or choline derivatives, they can provide a cytoplasm-like natural environment, meaning that enzymes can act in a natural environment. As reported by Veerporte (ref. 1) DES have been proposed to represent the real natural environment in which physiological processes take place.

  1. On page 20, line 627, the correct word is "fed-batch".

The mistake has been corrected.

  1. Please check the references. For example, in reference 37 the journal is not mentioned.

All the references have been carefully checked and, where the case, corrected.

Reviewer 2 Report

This review lays out the latest updates on the use of DES in biocatalytic reactions. The above contents were systematically reviewed with reductions, oxidations, hydrolyses, esterifications, transesterifications and other methods as classification criteria. The literature review were interesting and valuable to the authors of Molecules. However, it should be clarified more clearly in the following questions: 

(1) Although the author has supplemented the latest (2010 2022) literature on the application of DES, it is necessary to point out whether the effect of DES in biocatalytic reaction has been improved or the method has been innovated compared with previous years.

(2) For some important application examples, the author should list some tables to compare their conversion or yield, so as to facilitate the comparison of various examples.

(3) The review considers reductions, oxidations, hydrolyses, esterifications and transesterifications, author should further compare the difference and relationship between the application of the combination of enzyme and DES in these reactions.

(4) As this is an approach for biocatalytic reactions, it is likely for certain limitations to be encountered. Do the authors have any of such limitations to report about DES? If yes, it should be reported and suggestions to remove this limitation can be recommended for future studies and make a detailed outlook in the conclusion.

(5) Please check the abbreviations in the text in detail, some of which are mentioned many times.

(6) The fonts in the table should be consistent, such as Table 1. Each table must have a brief title that describes the contents. The title should be understandable without reference to the text. Therefore, the captions of tables in the text are too simple and should be described in more detail.

In conclusion, the paper is systematic and interesting, it could be published with minor changes.

Author Response

Reviewer n. 2

This review lays out the latest updates on the use of DES in biocatalytic reactions. The above contents were systematically reviewed with reductions, oxidations, hydrolyses, esterifications, transesterifications and other methods as classification criteria. The literature review were interesting and valuable to the authors of Molecules. However, it should be clarified more clearly in the following questions: 

  1. Although the author has supplemented the latest (2010 − 2022) literature on the application of DES, it is necessary to point out whether the effect of DES in biocatalytic reaction has been improved or the method has been innovated compared with previous years.

The need for an update on the subject is testified by the increasing number of references that appeared in the literature in the last decade and are reported in the review. Not only are there new methodologies but also the use of DES in biocatalytic transformations has been extended to new classes of enzymes.

  1. For some important application examples, the author should list some tables to compare their conversion or yield, so as to facilitate the comparison of various examples.

The Tables in the review already report yield and conversion for each example, in order to have a clear view of the advantages of this combination.

  1.  The review considers reductions, oxidations, hydrolyses, esterifications and transesterifications, author should further compare the difference and relationship between the application of the combination of enzyme and DES in these reactions.

The review is in itself a selection of all those situations in which the application of the combination DES/enzyme brought an increase in yield or enantioselectivity. In most of the examples cited the reported reaction is compared with the results obtained in standard PBS.

  1. As this is an approach for biocatalytic reactions, it is likely for certain limitations to be encountered. Do the authors have any of such limitations to report about DES? If yes, it should be reported and suggestions to remove this limitation can be recommended for future studies and make a detailed outlook in the conclusion.

We thank the referee for the suggestion. We added a paragraph on the possible limitations in using  DES in the conclusion.

  1. Please check the abbreviations in the text in detail, some of which are mentioned many times.

We checked the abbreviation all through the manuscript and added their meaning when missing.

  1. The fonts in the table should be consistent, such as Table 1. Each table must have a brief title that describes the contents. The title should be understandable without reference to the text. Therefore, the captions of tables in the text are too simple and should be described in more detail.

The Tables have been changed in order to meet the referee’s criticism.

In conclusion, the paper is systematic and interesting, it could be published with minor changes.

Reviewer 3 Report

 This review manuscript reports interesting information on the field of promoting organic synthetic reactions, through the use of deep eutectic solvents. Nevertheless, this article focuses on the enzymatic reactions, it reviews systematically important biotechnological reactions and processes, and thus this work gets of a specific importance; many significant enzymatic reactions are cited.

  Therefore, the reader concludes easily that the use of deep eutectic solvents and/or the natural deep eutectic solvents are the potential sustainable alternative (versus ionic liquids) to avoid known drawbacks of organic solvents in enzymatic synthesis, including the environmental risks. Moreover, the whole-cell catalysis is also reviewed.

Strength points of the manuscript:

(1) Well Structured

(a) Abstract,

(b) Introduction,

(c)Reductions (Reactions with isolated enzymes / Tables comprising example reaction schemes with isolated enzymes, and whole-cells reductions, are included),

(d) Oxidations (Oxidation reactions with isolated enzymes / Tables comprising example reaction schemes with isolated enzymes, and whole-cells oxidations, as well as cases of enzyme catalyzed reactions by specific oxidases, are included),

(e) Hydrolysis (Hydrolytic reactions with isolated enzymes / Tables comprising example reaction schemes with isolated enzymes, and whole-cells hydrolyses, are included),

(f) Esterifications and transesterification (Enzymatic esterification and transesterification reactions with isolated enzymes / Tables comprising example reaction schemes with isolated enzymes are included),

(g) Miscellaneous reactions (Enzymatic miscellaneous reactions / Tables comprising example reaction schemes are included),

(h) Conclusions (thoroughly)

(i) References (125)

(2) Comprehensive text

(3) A plethora of example reactions are given within extensive Tables

(4) All tabulated example reactions are important in the organic synthesis and biotechnological applications

Weakness points of the manuscript:

(1) The text needs of a careful English language syntax corrections

  Oveall, I can recoomend the publication of this review in the Journal molecules. Moderate English changes are required

Author Response

Referee n. 3

We thank the referee for his/her nice comments and appreciation.

The text has been rechecked for the English language and the readability improved.

Round 2

Reviewer 1 Report

I recommend the publication of the article.